# Effectiveness of distributed temperature measurements for early detection of piping in river embankments

Silvia Bersan[1], André R. Koelewijn[2], Paolo Simonini[1]

[1] Department of Civil, Environmental and Architectural Engineering, University of Padova, Padova, 35131, Italy
[2] Department of Dike Technology, Deltares, Delft, 2600 MH, the Netherlands.

*Correspondence to*: Silvia Bersan (silvia.bersan@dicea.unipd.it)

**Abstract.** Internal erosion is the cause of a significant percentage of failure and incidents involving both dams and river embankments in many countries. In the past 20 years the use of fibre-optic Distributed Temperature Sensing (DTS) in dams has proved to be an effective tool for detection of leakages and internal erosion. This work investigates the effectiveness of DTS for dike monitoring, focusing on early detection of backward erosion piping, a mechanism that affects the foundation layer of structures resting on permeable, sandy soils. The paper presents data from a piping test performed on a large-scale experimental dike equipped with a DTS system together with a large number of accompanying sensors. The effect of seepage and piping on the temperature field is analysed, eventually identifying the processes that cause the onset of thermal anomalies around piping channels and thus enable their early detection. Making use of dimensional analysis, the factors that influence that thermal response of a dike foundation are identified. Finally some tools are provided that can be helpful for the design of monitoring systems and for the interpretation of temperature data.

**Keywords:** Distribute Temperature Sensing, fibre optics, internal erosion, backward erosion piping, levee monitoring

# 1 Introduction

## 1.1 Backward erosion piping

Backward erosion piping is a specific kind of internal erosion mostly occurring under water retaining structures that are founded on fine to medium sand (ICOLD, 2015). Soil particles are removed and transported by seepage flow; as a consequence a thin pipe is formed below a roof provided by a layer of cohesive soil or a rigid structure. The erosion process typically starts at the downstream side of the structure, where the flow lines converge at an unfiltered exit (Figure 1). The erosion channels, or pipes, grow backwards and when they reach the upstream side a pressure surge can cause an excessive enlargement often followed by the failure of the embankment. Backward erosion piping can also occur, although less frequently, under revetments (Galiana, 2005) or in riverbanks and dike bodies with sandy inclusions (Hagerty, 1991).

Backward erosion piping represents a significant problem for the safety of river and sea dikes[1] located in delta areas, including, for example, the Po plain in Italy and large part of the Netherlands.

A number of studies have been conducted to describe the piping process and define failure criteria. The first studies date back to the beginning of the last century (Clibborn and Beresford, 1902; Bligh, 1910; Terzaghi, 1922) and still partially form the basis of the rules included in national codes (USACE, 2005). Many other experimental and theoretical studies followed across the years (e.g. WES, 1956; De Wit et al., 1981; Weijers and Sellmeijer, 1993; Muller-Kirchenbauer et al., 1993; Schmertmann, 2000; Ojha and Singh, 2003; Sellmeijer et al., 2011; Van Beek et al., 2011). Later studies show that researchers are still investigating the subject to improve the theoretical model (e.g. Richards and Reddy, 2014; Van Beek et al., 2014a,b; Allan et al., 2014) and find cost-effective countermeasures (Koelewijn et al., 2014; Wang et al., 2015).

## 1.2 Monitoring piping[2]

Assessing the safety level of existing dikes for piping is quite challenging because of the uncertainty associated with the theoretical model and, even more importantly, the uncertainty related to the spatial variability in the subsoil composition. The latter is favoured by the large extent of the stretches to assess and the geology typical of river environments. As a matter of fact, ancient riverbeds and crevasse channels are likely to cross the current course of the rivers causing a local change in the type of sediments lying under the dikes. Monitoring systems help identifying the zones where stronger seepage is recorded, which often represents weak spots where failure is more likely to occur. Identification of the most hazardous stretches along a dike is the first step towards a rational allocation of the resources for both inspection activities and improvement works.

Monitoring systems can also work as early warning tools. In this case their effectiveness depends on how much in advance they can detect the incipit of failure in comparison with the time necessary to undertake and complete the actions aimed at

---

[1] Throughout the text mostly *river* dikes are mentioned, but the statements and findings can also be applied to sea dikes.

[2] For the sake of brevity the term *piping* is used in this paper with the meaning of backward erosion piping. For indicating internal erosion in general (including contact erosion, concentrated erosion, suffusion etc.) the term *internal erosion* is used.

preventing the collapse or to complete the evacuation of the protected areas. In river embankments, a few sudden failures promoted by piping have been observed (e.g. Imre et al., 2015) but in most cases the erosion process is slow enough to allow effective intervention, which consists in building temporary sandbag rings around the sand-boils to stop or slow down the sand transport. The double possibility of either fast or slow evolution of the mechanism finds confirmation in theory (Van Beek et al., 2014a) and experiments (Koelewijn and Taccari, 2017).

Identification of piping is generally performed through visual inspections during flood events. The main limitations related to this approach are the lack of qualified personnel and the fact that sand-boils can be difficult to spot. As a matter of fact, they can be hidden by vegetation or by ponding water at the embankment toe, they can develop in ditches or hundreds of meters away from the dike toe. In this context sensors represent "extra eyes" with the capability of extending the control domain beyond what is visible and anticipating the detection of anomalies (Peeters et al., 2013).

Piping is also hard to identify using conventional geotechnical instrumentation such as pore pressure sensors or inclinometers, because its effect on porewater pressures is highly localised and because often no significant deformation is produced before the collapse of the structure. Both theoretical considerations (Ng & Oswalt, 2010) and field data from the test described here (Bersan, 2015) show that the draining effect of a pipe affects a region around the pipe with a radius in the order of 1 m. Consequently, a monitoring system relying on measurements of porewater pressure would require a high, and therefore uneconomical, number of sensors, even for controlling short stretches.

In the last decades, many efforts have been made to identify new parameters which are linked to seepage and internal erosion and can be measured by means of extensive technologies. Amongst these are electrical resistivity, self-potential and temperature (Sheffer et al., 2009). The common shortcoming of these methods is that they measure quantities that are influenced by a number of variables besides the occurrence of internal erosion, which makes data interpretation far from straightforward. As an example, see the inversion of self potential data that Rittgers et al. (2015) performed for an experiment similar to the one described in this paper. Affected by smaller uncertainties than electrical resistivity and self-potential, the thermometric method has gained increasing acceptance for detection of leakages and internal erosion in the past two decades (Johansson & Sjödahl, 2009).

## 1.3 The thermometric method

The method is based on the principle that, in the absence of seepage (or in presence of moderate seepage flow), the temperature distribution in the upper portion of the subsoil fluctuates seasonally in response to the variations of the air temperature. Daily variations of the air temperature propagate about 1 m below the surface, while annual variations propagate up to 10-15 m depth. Further deep the temperature is constant and mildly increases with depth with an average gradient of 25 °C/km. When pore water flows at significant velocity the transfer of heat promoted by the fluid flow, denominated *advection*, becomes predominant over the heat conducted from the surface. Therefore, sections of dam or dike characterized by markedly different seepage regimes are expected to present different temperature distributions. Since internal erosion creates zones of higher

permeability and, consequently, a localized increase of the seepage rate, it is expected to causes a local variation of the temperature field.

In the past temperature measurements were performed at discrete locations using thermocouples installed at different depths inside standpipes. After the introduction of Distributed Temperature Sensing (DTS) systems, it became possible to perform continuous measurements over kilometres with a spatial resolution down to 1 m. Optical cables can also be installed along existing embankments. Easily accessible locations are the landside toe on the landside slope. Details on the functioning principles of fibre-optic sensors can be found, for instance, in Henault et al. (2010). There are two variations of the DTS technique. The passive method, that is the method adopted in this work, consists in measuring the temperature variations naturally occurring in the soil. The active method, or heat pulse method, consists in heating the cable containing the optical fibres and observing how it cools down: higher seepage flows correspond to a faster cooling.

To date DTS systems have been installed in concrete dams (in the foundation soil), earth dams, concrete-face rock-fill (CFRF) dams (to detect leakage from joints), waterproofed basins and canal dikes (Johansson and Sjödahl, 2004; Thongthamchart, 2012; Smartec SA, 2017; Courivaud et al., 2011). Very few installations, however, concern river or sea dikes. The cost of the data acquisition unit could be one of the factors limiting widespread use of these sensors. Nevertheless, the diffusion of DTS systems in many other fields has led to a significant decrease of their costs in the past decade and a further reduction is expected in the coming years, making the technique appealing even for applications where the economic investments are smaller, such as in river management.

Several studies concerning the influence of internal erosion on the temperature field of damming structures have been conducted (Johansson, 1997; Guidoux, 2008; Radzicki and Bonelli, 2010a; Cunat, 2012) and different data interpretation techniques have been developed. First interpretation techniques (Claesson at al., 2001) were developed for dams and were suitable for cases with constant hydraulic loads and sensors located deep enough not to be influenced by the periodical variations of the external temperature. Later on, data interpretation tools were designed for cases where the sensor is located close to the surface so that the temperature recorded is strongly influenced by the external environment (Radzicki and Bonelli, 2010b; Khan et al., 2008, 2010).

The thermometric method has also been extensively exploited to study exchange fluxes between groundwater and surface water within streams, lakes and wetlands. Data are collected using couples of thermal loggers installed at two different depths along a vertical. One-dimensional analytical solutions are typically used to infer vertical fluxes from the amplitude ratio or the phase shift of the diurnal temperature signal between pairs of sensors located at different depth (Irvine et al., 2017; Briggs at al., 2014). Fibre-optic distributed temperature sensors have been installed along streambeds and lakebeds to gain information on the spatial variability of the groundwater-surface water exchange fluxes (Krause at al., 2012; Tristam et al., 2015). The adoption of distributed temperature sensors has also been proposed for quantification of groundwater fluxes; for this purpose pairs of optical cables must be installed parallel to each other with a small vertical separation (Becker et al., 2013; Mamer and Lowry, 2013).

#### 1.4 Aim and objectives

In this work the effectiveness of DTS to detect backward erosion piping occurring in the foundation of river and sea dikes is discussed. Although the use of DTS for early detection of internal erosion is nowadays quite common, specific studies are advisable to verify the suitability of the technique for river embankments and to assess what the optimal solutions are in terms of sensor layout and data analysis. As a matter of fact, there are important differences between dams and river dikes which influence the thermal response of the structure and might have an influence on the effectiveness of the thermometric method. The main differences are the size of the embankment and the duration of the hydraulic loads, nearly constant in dams while persisting from 12 hours to a few days in river and sea dikes.

This paper illustrates the response of a distributed temperature sensor installed in a test dike in which piping was induced by gradually increasing the hydraulic load over a 5-day period. Based on the evidence provided by the data different processes that can lead to the onset of thermal anomalies in regions affected by piping are identified. Making use of the theory of heat transfer in hydrogeological systems the conditions required for the onset of thermal anomalies are discussed. As a result, a conceptual model of the operation of DTS for piping detection in river dikes is formulated, which can be helpful for the design of monitoring systems as well as for basic data interpretation. It could also be the basis for the development of more complex data interpretation tools.

#### 2 Theory and methods

#### 2.1 Theory of heat transfer in hydrogeological systems

Heat transfer in porous media is described by the following form of the advection-diffusion equation:

$$\frac{\partial T}{\partial t} + \nabla \cdot \left( \frac{\rho_w c_w}{C} \mathbf{u} T \right) = \nabla \cdot \left( \frac{\lambda}{C} \nabla T \right) \qquad (1)$$

where $T$ is the temperature, $\rho_w$ and $c_w$ are density and specific heat capacity of water, $C$ is the volumetric heat capacity of the soil, $\lambda$ the thermal conductivity of the soil. Typical values of the thermal properties are given in Table 1; these were used for the analysis presented in the following.

The second term of the left-hand side is the advective term and describes the transport of heat operated by the fluid moving in the pores. $\mathbf{u}$ is the Darcy velocity, which is given by the Darcy's formula:

$$\mathbf{u} = -K \, \nabla \left( \frac{p}{\rho_w g} + z^* \right). \qquad (2)$$

In Eq. (2) $K$ is the hydraulic conductivity, $\rho_w$ is the density of water, $p$ is the porewater pressure, $z^*$ is the elevation and $g$ is the gravitational constant. The quantity

$$\mathbf{v}_T = \frac{\rho_w c_w}{C} \mathbf{u} \qquad (3)$$

is usually referred to as the thermal front velocity. Although it is not the exact velocity at which a thermal front advances, it is considered to be an accurate measure of the front velocity.

The right-hand side of Eq. (1) is the diffusive term that describes the effect of heat conduction. The quantity

$$a = \frac{\lambda}{c} \qquad (4)$$

is called thermal diffusivity and is a measure of the inertia of the medium to temperature changes.

Eq. (1) neglects thermal dispersion, which is the spreading of heat caused by the fluctuation of the micro-streamlines with respect to the main fluid flow direction because of the existence of a pore system. Although in solute transport - which is described by the same equation - dispersion is significant, Rau et al. (2012) show that, at the sub-meter scale, dispersion of heat in soils is negligible. Dispersion can however occur at a larger scale because of the combined variability of micro-streamlines and hydraulic conductivity.

In absence of groundwater flow ($u = 0$), the temperature distribution in the subsoil can be estimated solving Eq. (1) in a 1-d domain. The model by Hillel (1998) assumes that the temperature at the soil surface ($z = 0$) has a sinusoidal variation throughout the year:

$$T(t) = T_a + A_0 \sin [ \omega(t - t_0) + \phi], \qquad (5)$$

where $T_a$ is the mean soil temperature, $A_0$ is the amplitude of the annual temperature function, $\omega = 2\pi/365$ is the angular speed, $t$ is the day of the year (starting counting from an arbitrary day). $t_0$ can assumed as the time lag between the occurrence of the minimum temperature in a year and the arbitrary reference day; in such case the phase shift is $-\pi/2$. The model also assumes that at infinite depth the soil temperature is constant and equal to $T_a$. The temperature at any depth $z$ is then a sine function of time and can be represented as:

$$T(z,t) = T_a + A_0 \, e^{-z/d} \left[ \sin \, \omega(t - t_0) - \frac{z}{d} + \phi \right]. \qquad (6)$$

The constant $d$ is a characteristic depth, called damping depth, at which the temperature amplitude decreases to the fraction 1/e of the amplitude at the soil surface. It is related to the thermal diffusivity of the soil and the frequency of the temperature fluctuation:

$$d = (2a/\omega)^{1/2}. \qquad (7)$$

When air temperature is used as input instead of surface temperature, the effects of solar radiation and wind convection are not taken into account correctly. Nofziger (2005) suggests increasing the input temperature of 2 °C to take into account the effect of solar radiation on bare soils; the correction should be smaller for vegetated soils.

In the presence of groundwater flow the temperature distribution in the subsoil depends on the flow velocity: if the velocity is relatively small the contribution of the advective term in Eq. (1) is negligible and the temperature distribution is again as given by Eq. (6). If the velocity is large the temperature field will be different. The relative importance of advection over conduction is described by the Péclet number, which can be derived from Eq. (1) by means of dimensional analysis. For a one-dimensional problem the Péclet number is defined as follows:

$$\mathrm{Pe} = \frac{u \, l \, \rho_w c_w}{\lambda}, \qquad (8)$$

where $l$ is the characteristic length, that is a length representative of the phenomenon under investigation. Exact threshold values for the Péclet number do not exist, but for values much less than unity conduction is dominant, whereas for values significantly greater than unity advection prevails; for values in the order of 1 the behaviour of the system is intermediate. Assuming a characteristic length of 10 m and typical values of the thermal properties of soils (see Table 1) the Péclet number is on the order of 1 when the Darcy velocity is on the order of $10^{-7}$ m/s.

A two-dimensional problem such as the transfer of heat in the foundation of a dike is more accurately described by a 2-d Peclet number. Van der Kamp and Bachu (1989) have defined a non-dimensional number, the geothermal Péclet number, which is specific for a hydrogeological system where the conductive heat flow is mainly vertical and the fluid flow is horizontal:

$$\text{Pe}_g = \frac{q_H \, \rho_w c_w \, D \, A}{\lambda} . \qquad (9)$$

In the above formula, $q_H$ is the average horizontal Darcy velocity, $D$ is the thickness of the hydrological system and $A = D/L$ is the aspect ratio of the representative element of the seepage domain, where $L$ is given by the horizontal size of the flow path. Conduction in the horizontal direction is neglected. Van der Kamp (1984) pointed out that Eq. (9) represents the ratio of the amount of horizontally convected heat (approximately equal to $q_H C_w D \Delta T$, where $\Delta T$ is the change in temperature between the top and bottom of the system) to the amount of heat transferred vertically by conduction (approximately equal to $\lambda L \Delta T / D$).

The most common discussion around Péclet numbers is about the choice of the characteristic length. In hydrogeological systems the characteristic length is related to the vertical size of the domain but the choice of its value is not straightforward. Van der Kamp and Bachu (1989) leave the question open for hydraulically non-homogeneous systems, i.e. systems where the fluid flow is concentrated in a layer of limited thickness compared to the depth of the system.

## 2.2 Piping test on a large-scale trial embankment

In September 2012, in Booneschans, in the Northeast of the Netherlands, backward erosion piping was induced in two trial embankments. The experiments were part of the IJkdijk (Dutch term for 'calibration dike') program, a research program initiated in 2005 with the double goal of testing new monitoring techniques under field conditions and advancing the knowledge on geotechnical failure mechanisms at large scale. The project involved research institutes, sensors manufacturers and water authorities. A previous similar experiment was carried out in 2009 and the related temperature data are presented in Artières et al. (2010) and Beck et al. (2010).

In this paper the test conducted on the 'west dike' is described. The dike is located on the right in Fig. 2. The test dike was surrounded by a containing ring forming a reservoir with a volume of about 2000 m$^3$. A lower and smaller dike ring enclosed a second basin used to control the downstream water level and ensured full saturation of the foundation layer. The latter was carefully prepared applying 50% vacuum during saturation.

The dike was 3.5 m high, 19 m long and 15 m wide at the bottom. The geometry of the dike is depicted in Fig. 3. The foundation consisted of a sand layer with a thickness of 3.25 m (3.00 m by design). An impermeable foil separated this layer from the in situ soil. The foundation soil was a uniform sand characterized by an average grain size of $d_{50} = 0.30$ mm and a uniformity

coefficient $U=d_{60}/d_{10}=1.69$. The lower part of the dike was made of a 0.7 m well-compacted clay layer. Given its low permeability, the clay layer separated the hydraulic fluxes occurring in the foundation from the fluxes affecting the embankment. The dike body was made formed by a 1.7 m high, poorly-compacted small clay dike on the upstream side and a sand core covered by organic clay. This composition is representative of a number of small dikes around the Netherlands.

The field trial was named All-In-One Sensor Validation Test, since the occurrence of more than one failure mechanism was possible. By design, the failure of the dike described in this paper could occur either because of piping through the foundation soil or because of micro-instability of the sand core. Failure by overtopping with subsequent erosion of the downstream slope was also a possibility, in case the previous two did not occur earlier.

Throughout the test, the dike was monitored using a number of innovative technologies with the aim of testing the capability
of such technologies in predicting imminent failure in field conditions. The technologies included distributed fibre-optic strain and temperature sensing, fibre Bragg grating (FBG), ground-based radar, ground penetrating radar and electrical resistivity. An infrared camera mapped the surface temperature of the downstream slope (and of part of the upstream and downstream basins).

The distributed fibre-optic sensor consisted of 2 single-mode and 2 multi-mode fibres encased in a geotextile strip (Fig. 4). A
single strip was arranged in 8 profiles along the length of the dike: five at the interface between the sand layer and the bottom of the dike (labelled F1 to F5 in Fig. 3), three on the downstream slope of the dike (labelled F6 to F8). Single-mode fibres were connected to a reading unit exploiting stimulated Brillouin scattering to measure strain while multi-mode fibres were connected to a reading unit exploiting Raman scattering to measure temperature. Temperature changes were measured with an accuracy of 0.1 °C, a spatial resolution of 1 m along the sensor and a frequency of two measurements per hour. According to the
laboratory calibration the accuracy of the measurements was higher than 0.1°C up to a fibre length of 1 km. Please note that in the field, the accuracy may be affected by additional installation issues not considered in laboratory calibration, such as sharp bends in the deployed cable, connections and splices, which may induce step losses not addressed by laboratory calibration (Tyler et al., 2009). In the field test, significant effort has been paid to avoid sharp bending: the cable has been deployed with curvature radii larger than 20 cm. Moreover, neither splices nor connectors were present in the section of fibre
under measurement. Nonetheless, splices and connectors were present elsewhere, and therefore a field calibration of the DTS system would have been advisable to compensate the corresponding step losses (Hausner et al., 2011; Hausner and Kobs, 2016). Unfortunately, field calibration was not carried out due to practical issues; for this reason some impairments on the overall accuracy of the system may be expected. Such impairments are expected to be of a rather small magnitude as the temperature variations expected over time are modest.

The dike was also equipped with conventional pore pressure transducers, which provided reference measurements and allowed the growth of the pipes to be closely monitored: 4 lines (labelled P1 to P4 in Fig. 3) consisting of 17 sensors each had been installed at the top of the foundation soil during construction and 3 lines (labelled P5 to P7) consisting of 3 sensors each had been installed in the sand core, right above the clay layer. The conventional monitoring also included two liquid level sensors

to record the water level in the upstream and downstream reservoirs, a flow meter placed at the discharge point of the downstream basin, visual inspection and hand readings of discharge and basin levels performed at regular intervals.

The dike was forced to collapse slowly, in a controlled manner, in order to provide the largest amount of data possible to aid in evaluating the performance of the sensors and understanding the ongoing mechanisms. The water level was increased in steps in the upstream basin, while the downstream basin was maintained at an almost constant level of about 10 cm above the surface of the sand layer to ensure full saturation. The resulting load (hydraulic head over the dike) is displayed in Fig. 5.

The bulk hydraulic conductivity of the sand layer, as calculated from flow measurements performed at the discharge point of the downstream basin before piping occurred, was $K = 1.5 \cdot 10^{-4}$ m/s.

## 3 Experimental results

### 3.1 Development of piping

After two days of testing ($t = 43$ h), at the hydraulic load of 1.50 m, two among the pore pressure sensors of line P1 (downstream) showed a small pressure drop. Two hours later the first sand-boil was detected at the location $x = 5.2$ m. At $t = 50$ h, at the hydraulic load of 1.75 m, three other pore pressure sensors in line P1 showed a small drop, later followed by the discovery of three new sand-boils at $x = 8.7$, 11.2 and 11.7 m. At $t = 55$ h, after the hydraulic load was increased to 1.85 m, pressure drops occurred in line P2, indicating that piping channels grew and reached that far. From that moment on, sand transport occurred continuously, as revealed by the size of the sand-boils increasing with time. Two more sand-boils were detected, at $x = 17.4$ m and $x = 7.0$ m, respectively at $t = 60$ h and at $t = 65$ h. After 67 h the opening of a controllable drainage tube that had been installed in the dike foundation as a countermeasure (its position is indicated in Fig. 3) caused a general pore pressure drop and the arrest of erosion. At 90 h visual inspection was stopped for safety reasons. At 94 h the drainage tube was closed to bring the dike to collapse.

During the test no significant pore pressure drop was recorded by the transducers at line P3. Therefore, it can be assumed that no pipe grew that far and no pipe reached the upstream side. Consequently, piping did not play any role in the collapse of the dike occurring on the 5th day of testing.

From the data collected during the field trial the size of the pipes can be roughly inferred. The length of the pipes in the first stages can be deduced knowing that the pipes had reached line P1 of pore pressure transducers, but not yet line P2. From the size of the sand-boils recorded during the visual inspections the volume of sand eroded is calculated and from these data the cross-sectional area of a pipe (or a multitude of pipes that developed behind a sand-boil) proves to be between 2.5 and 5 cm$^2$.

### 3.2 Temperature data

Temperature measurements started 7 days before the beginning of the test. Fig. 6 shows the evolution of the temperature measured at lines F1 to F5 at the middle section of the dike ($x = 10$ m). From the beginning of the measurements ($t = 0$) to the

beginning of the test ($t$ = 7 days), the temperature at the bottom of the dike was nearly constant. Minor differences, smaller than 1 °C, were due to different distances of the measuring lines from the surface and different expositions (south or north) of the slope under which the point is located. The only exception is represented by line F1. Being the shallowest and lying under the slope facing south, in absence of water flow line F1 was the most influenced by the external environment. Its temperature increased nearly monotonically until the beginning of the test as a consequence of the increase in the average daily temperature and solar radiation during the week preceding the test. Since there is always a time shift between the trend of the air temperature and the trend of the soil temperature, in the 2 days preceding the beginning of the test the soil temperature kept increasing although the average air temperature was dropping. Immediately after the beginning of the test the temperature at F1 started decreasing since the effect of the external temperature was completely masked by the effect of seepage.

A few hours after the beginning of the test, the temperature recorded at F5 started increasing, suggesting that the reservoir water was warmer than the foundation bulk. Unfortunately, measurements of the water temperature in the upstream basin were not available due to technical problems. At lines F4 and F3 the warm front arrived after 2 and 3 days respectively. On the contrary, at F2 the temperature started to slowly decrease soon after the beginning of the test and kept decreasing until the arrival of the warm front, more than 3 days later. At line F1 the temperature kept decreasing during the test and the warm front never arrived.

The spectrogram in Fig. 7 depicts the temperature measured at the most downstream line (F1) as a function of time. Starting from 55 h, localized temperature drops are visible at the locations where piping was observed, approximately at $x$ = 5, 11 and 17 m. The time coincides with the recording of the first pressure drops at line P2. No anomaly in the temperature is detected at 43 h, when the first pressure drops where recorded at the downstream line P1. This is consistent with the position of F1, which is located between P1 and P2. The local temperature variations measured during the test were all smaller than 1 °C. No temperature anomaly was observed along the lines F2 to F5.

The mechanism that led to the formation of the thermal anomalies is clarified by Fig. 8, which shows a plan view of the temperature at base of the dike at four successive times during the test. The first contour (t=0) again shows that the initial temperature was significantly higher under the toe of south slope (y=15 m). The sequence then shows that the warm inflowing water preserved its initial temperature while advancing under the dike. The advancing warm water also pushed the water initially present under the dike crest progressively downstream, promoting the advance of a cold thermal front towards the (initially) warmer downstream toe. The cold front advanced faster at the piping locations than in the soil unaffected by piping, thus producing the measured localized temperature drops. In the 100 h contour plot, preferential flow paths can be observed at the sides. Here warm water flows faster than in the centre of the dike. As no sand-boil was observed at the sides, these preferential paths are believed to be a consequence of the discontinuity between the soil and the impermeable foil delimiting the artificial basin. These leakages could not be identified by the pore pressure readings. The contours also show that the temperature is not uniform along the x-axis (excluding downstream), neither in absence of flow ($t \leq 0$) nor with seepage occurring ($t > 0$). The temperature is indeed higher at the centre than at the borders. This can be ascribed to the 3-d character of the test facility, unlike a real dike where the longitudinal dimension is much larger than the transversal dimension.

As suggested by Khan (2008), temperature gradients in the longitudinal direction can be very informative for detection of anomalies. Fig. 9 shows the gradients along the fibre F1, calculated as the temperature difference between two points along the fibre located 1 m apart. At locations x = 5 m and x = 11 m, where sand-boils have been observed, the gradients start to increase at 50 h, stabilize around a value of 0.3 between about 65 and 85 h, when the hydraulic load is kept nearly constant and then decrease after 85 h, when the hydraulic load is increased at a rate of roughly 12 cm per hour. At x = 18 m, where a sand-boil was observed starting from 60 h, the large magnitude of the spatial temperature gradient is likely to derive from the superposition of two effects: the temperature decrease caused by piping at that location and the temperature increase caused by the leakage along the foil at $x = 19$ m (cf. Fig. 8). For comparison, the graph also reports the gradients at $x = 14$ m where no sand-boil was observed, at least not until 90 h when visual inspection was stopped. In real embankments high temperature gradients along the dike toe can be the expression of spatial variability in the hydraulic conductivity of the foundation soil rather than of internal erosion. Spots of higher conductivity are anyhow interesting to map because they represent preferential flow paths where the risk that piping occurs is higher.

Fig. 10 shows the temperature gradient in the seepage (transverse) direction, calculated in proximity of the downstream toe, between fibre F1 and F2, at two locations: $x = 11$ m, where piping occurred, and $x = 14$, where no trace of piping was detected. The gradient is non-zero at the beginning of the test and keeps decreasing during the test as a consequence of the seepage flow. After 50 h the gradient decreases faster at the location affected by piping. Here, after 90 h the gradient turns negative. Indeed, towards the end of the test, the cold water that was initially under the crest is now at the downstream toe (F1) and the warmer water initially under the upstream toe is at F2. The even warmer reservoir water could also have reached F2.

The temperature anomalies detected were of very small extent, in the order of few tenths of a degree. Since the spatial resolution of the sensor is much larger than the width of a single pipe, it was initially feared that the averaging performed by the sensor was reducing the actual magnitude of the temperature anomalies. However, when the pipes developed, the temperature difference between F1 and F2, which was responsible for the anomalies produced by piping, was less than 1 °C and thus the small extent of the anomalies was largely to be ascribed to the field conditions rather than to the measuring system. The possibility that a branched net of very small pipes formed rather than a single larger channel, as observed in medium-scale experiments (Weijers and Sellmeijer, 1993) would translate in a wider eroded area and a reduced impact of the resolution of the sensor on the measured temperature anomaly.

The small extent of the anomalies could also raise concern about the impact that the accuracy of the sensor has on the results. As previously stated, according to the laboratory calibration the accuracy of the measurements was higher than 0.1°C, but in the field the accuracy may be affected by step losses occurring at sharp bends, connections and splices along the cable. Since the main interest here is in relative measurements, i.e. temperature variations in space and in time, significant influence of step losses can be excluded because neither splicings nor connectors were present in the section of fibre installed under the dike. Moreover, if we consider the most informative part of the sensor, i.e. the most downstream line F1, also step losses induced by bends can be reasonably excluded. Of course all the step losses occurring between the reading unit and the line F1 could affect the accuracy of the measurements performed at the latter, but their effect on the relative measurements is expected to be

negligible. It can be also excluded that the temperature measurements were influenced by excessive strain experienced by the fibre during the experiment. Indeed, the maximum strain measured by the single-mode fibre after four days of testing was around 400 µstrain. In addition, the strain experienced by the multimode fibres dedicated to temperature measurements was certainly much smaller since these fibres are placed into loose tubes to prevent excessive stresses.

## 4 Conceptual model

### 4.1 How DTS can detect backward erosion piping

Interpretative schemes devised for temperature measurements performed in earth dams assume that heat transfer occurs mostly by advection in the zones affected by internal erosion and by conduction in the rest of the soil (Johansson and Hellström 2001; Johansson and Sjödal 2009). Such schemes well apply to low permeability bodies such as dam cores. They also assume that the eroded zone extends from the waterside to the landside. Moreover, it is often stated that the existence of a temperature gradient between the waterbody and the soil is a necessary condition for the effectiveness of the temperature measurements (hence the name *gradient method,* which is sometimes used in lieu of passive thermometric method).

During the experiment described above the seepage flow induced significant temperature variations in the entire dike foundation, meaning that advection prevailed not only in the zones affected by internal erosion but everywhere in the foundation soil. In addition, the temperature of the waterbody did not have any influence on the formation of the thermal anomalies. The first outcome of the test is therefore a demonstration that the conceptual model commonly used for dams cannot be applied tout court to river embankments prone to piping.

The effect of soil permeability, scale of the structure and length of a backward eroding cavity on the development of thermal anomalies in damming structures have been investigated by Radzicki and Bonelli (2010a). In this paper complementary results are presented that have been obtained focussing on the mechanism of backward erosion piping and on the effect of hydraulic loads of short duration.

A conceptual model is first presented. It takes into account the necessity to detect piping in its early stage, that is when pipes are still confined close to the landside toe of the embankment. The main assumption of the model is that a sufficient condition for the formation of a thermal anomaly is a difference between the soil temperature at the *head* of the pipe (Fig. 1), that is where a pipe has its major intake, and the soil temperature at the downstream toe of the dike, where the sensor is commonly placed. The large-scale experiment demonstrated that such temperature difference can occur as the consequence of (at least) two different processes.

**Process 1: Riverside advective front.** The first process consists in the propagation of an advective thermal front from the riverside toe up to the head of the pipe (or slightly behind it). The water that enters the pipe then flows fast to the toe of the embankment and a distributed sensor located there registers a localized temperature drop or increase compared to the other regions that have not yet been reached by the advective front. This process requires that three conditions be satisfied: (i) the temperature of the waterbody is significantly different from the subsurface temperature at the point where the sensor is placed;

(ii) advection prevails over the distance between the riverside toe and the head of the pipe; (iii) the thermal front reaches the head of the pipe in a relatively short time compared to the time required for the full development of piping. Condition (i) is sometimes not satisfied during the mid-seasons when it is likely that both the waterbody and the soil approach the annual average temperature. The other conditions depend on a number of features such as geometry, hydraulic and thermal soil properties, magnitude of the hydraulic load and are discussed in Sec. 5.1.

**Process 2: Initial gradient.** Temperature anomalies can also arise regardless of the temperature of the waterbody. This second process occurs when - at the beginning of a flood event - the temperature under the dike is non-uniform and a temperature gradient exists along the seepage path that will be named here *initial gradient*. In low permeability soils, where conduction prevails, during a flood event the initial (non-homogeneous) temperature distribution is altered only in the pipes. As shown in the experiment, in more permeable materials the initial temperature distribution is altered everywhere by seepage, but in the pipes it is modified more quickly than in the unaffected regions. The initial gradient is generated by the conductive heat fluxes between the soil and the atmosphere. Such gradient can only develop if no seepage occurs under the dike except during flood events or in loosing/gaining streams if seepage velocity is small enough that conduction prevails over advection or conduction and advection have similar magnitude. The initial gradient is further discussed in Sec. 5.2.

**Combination of the processes.** The two processes can both occur in the same dike during the same flood event. The second process is exploitable for detection from the beginning of a flood event but tends to diminish its effectiveness with time. This was observed in the experiment, where the strength of the thermal anomalies - expressed in terms of temperature gradients along the dike toe - decreases towards the end of the test although it is unlikely that piping had stopped (Fig. 9). Since the temperature at the base of a dike is pretty uniform far from the toe, once that the water initially under the toe has been pushed forward and the water initially under the crest has taken its place, the temperature at the head and tail of the pipes is equal. On the contrary, the first process is effective only after some time from the beginning of the flow. The experimental data suggest that the riverside thermal front could have reached the head of the pipe close to the end of the test. The two processes can therefore act in synergy, one after the other, allowing continuous detection of piping.

The experimental data also suggest that a decrease of the magnitude of a thermal anomaly cannot be automatically interpreted as a decrease of the magnitude of the leakage and that for dikes subject to transient hydraulic loads it is very difficult to define a unique relationship between thermal anomalies and leakage magnitude, as it has been done for channel dikes (Artiéres et al., 2007).

## 4.2 Optimal position of the sensor

The experimental results confirmed that the optimal location of a DTS sensor for detection of backward erosion piping is close to the landside toe of the embankment. As a matter of fact only the most downstream portion of the sensor (F1, Fig. 3) could detect localized temperature variations that could be associated with piping.

One among the main differences between the experiment and a real installation is that the test dike was built on a homogenous foundation and the sensor was installed before the impervious base of the dike was placed; the sensor was therefore located

exactly at the depth were piping developed. In existing dikes this is difficult to achieve. The possibility to detect piping if the sensor is not located exactly where piping occurs are discussed with the help of the flow net illustrated in Fig. 11, obtained as result of a numerical model described in Bersan et al. (2013). Although most of the water enters a pipe in proximity of its head, a modification of the flow field is induced in all the surroundings of the pipe. The flow lines bend towards the pipe and the

vertical component of the seepage flow becomes quite significant at the bottom of the pipe. An upward flow can modify the initial temperature gradient normally existing in the vertical direction, producing a thermal anomaly not only inside the pipe or in a very small region around it (where heat is conducted from the pipe to the soil) but also tens of centimetres under the pipe. The vertical flow induced by the pipe is however more pronounced near the head of the pipe and decreases towards the tail, which suggests that the optimal position of the sensor is slightly behind the landside toe.

Especially when the foundation soil is heterogeneous or the depth of the interface between the sand layer and the above impervious layer is highly variable we suggest deploying the sensor in more levels at different depths and maintaining the depth of each level constant along the dike. If the sensor is installed at variable depth along the dike following the interface between layers, it is important to map its position very accurately to facilitate interpretation of the data.

The experiment also showed that it can be very helpful for the interpretation of distributed temperature data if measurements

are available also for a few points along the seepage path. This information can come for example from a couple of cross-sections instrumented with multiple piezometers each one including a thermal sensor. Such layout has recently been implemented in a dike stretch along the Adige river in Italy. The idea of a 3-D thermal monitoring system, although with some differences, is also proposed by Radzicki et al. (2015).

## 5 Numerical modelling and dimensional analysis

**5.1 Predicting the propagation of an advective front under a dike**

The evolution of the temperature under the test dike was simulated using COMSOL Multiphysics®. The equations describing heat transfer and seepage flow were solved in a coupled manner. Details on the numerical model are available in Bersan (2015). The model was first calibrated using the pressure and temperature measurements from the piping test, then used to predict the behavior of the test dike over a period longer than the test duration, in order to understand the effect of the time factor on the

thermal response of a dike. Simplified boundary conditions were assumed compared to the field conditions: uniform initial temperature of 12 °C, constant inflow temperature of 16 °C and constant hydraulic load of 3 m. The latter corresponds to an average seepage velocity of $3 \cdot 10^{-5}$ m/s. The results in Fig. 12 show that eventually the advective warm front reaches the landside toe, but this takes more than four days. It is inferred that in a small dike with a permeable foundation a seepage flow strong enough to induce piping can propagate an advective front from the riverside up to the landside toe, provided that the

hydraulic load is sustained for a long time. However, many levees are subject to significant hydraulic loads for very short periods, in the order of half a day to a few days. The time required for the thermal front to reach the downstream toe can therefore be of the same order or longer than the duration of typical flood events.

In order to extend the results of the experiment to a wide range of cases dimensional analysis was used. The applicability of the geothermal Péclet number in Eq. (9) to our problem was assessed with the help of a numerical model again developed in COMSOL Multiphysics® and solving Eq. (1). The model approximated the foundation by a rectangular domain as shown in Fig.13. The same temperature was assigned at the top, bottom and outflow boundaries; a different temperature was assigned at the inflow. Modelling only the sand layer and neglecting the convective fluxes occurring above and under the layer itself is a strong simplification that produces overestimation of the vertical conductive flux. However we considered it acceptable since our purpose was determining ranges rather than exact values. The choice was also motivated by the difficulty anyhow encountered in correctly placing the bottom boundary. A parametric study in steady-state conditions was conducted varying the length and thickness of the domain as well as the seepage velocity. It emerged that the geothermal Péclet number well describes the behavior of the system only if $D$ is substituted by $D/2$ in Eq. (9). The results are presented in Fig. 14: for values of the geothermal Péclet number much smaller than 1 the system is conduction-controlled, for values between 1 and 10 the behaviour is intermediate whereas for values larger than 10 the system is advection-controlled. The choice of substituting $D$ is with $D/2$ can be explained physically the fact that in the model by Van der Kamp and Bachu (1989) conduction occurs in a single direction, promoted by the temperature difference between the surface and the bottom of the hydrogeological system, whereas in our model conduction occurs from the centre of the domain (at temperature $T_w$) both upwards and downwards (both at temperature $T_0$).

In Fig. 15 the geothermal Péclet number is plotted as a function of $D$, the thickness of the sand layer, and $L$, the distance along the seepage path. In order to know whether a thermal front will propagate from the riverside up to the landside toe, $L$ must be chosen equal to the width of the dike at the base. Choosing smaller values, the propagation of the front can be predicted all along the seepage path. Two cases are plotted in Fig. 15: a very permeable sand layer (hydraulic conductivity on the order of $10^{-3}$ m/s) and a sand layer of medium permeability (conductivity on the order of $10^{-4}$ m/s). In both cases a uniform horizontal gradient $i$ of 0.1 is considered. The time required by the thermal front to reach the downstream toe is estimated using the thermal front velocity in Eq. (3) and can be read on the top $x$-axis as a function of $L$. This estimate represents an upper bound, since the effect of conduction is neglected, but the approximation is reasonably good. For the soil of high hydraulic conductivity the expected behaviour is intermediate or advective for every possible size of the dike and a thickness of the sand layer larger than 1. The time required for the front to reach downstream is in the order of a few days. For $K = 10^{-4}$ m/s the expected behaviour is purely conductive if the sand layer is thinner than 3-4 m. However, the main constraint seems to be the time required for the front to reach downstream, which is in the order of tens of days for a medium/large dike.

**5.2 Effect of initial temperature distribution on piping detection**

A temperature gradient is generally present at the base of a dike because of its geometry: in every slope the distance between the base and the surface increases from the toes to the top; an increasing distance from the surface translates in an increasing

phase shift and damping of the temperature wave propagating in the soil. Fig. 16 shows an example of the temperature distribution that can occur at the base of a 1:2 slope.

The temperature distribution in Fig. 16 was calculated using the finite element software COMSOL Multiphysics® to solve Eq. (1) under the hypothesis that $u = 0$. The same result would be obtained for small seepage velocities. Eq. (5) was applied as boundary condition at the surface of the slope with the following parameters: $T_a = 9.0\ °C$, $A_0 = 8.5\ °C$, $t_0 = 20$ d. The bottom boundary was located at the depth of 20 m from the base of the slope; there, a constant temperature value equal to the annual average $T_a$ was assigned. We used the numerical model to investigate the limitations related to the detection of backward erosion piping when the initial temperature gradient is exploited.

Fig. 17 shows the gradient along the base of an embankment slope. It can be observed that even in mid-seasons the gradient is sufficiently large to allow detection of a pipe just 1 m long with the temperature resolution commonly offered by DTS systems (provided that the pipe exactly crosses the sensor). The sensitivity of the model to soil thermal properties was tested: the dotted lines show that decreasing the thermal conductivity and heat capacity of the embankment and upper layer soil (which means assuming higher clay content and/or lower degree of saturation) the gradient slightly increases close to the toe, which means there is an even higher probability to detect piping.

From the graph in Fig. 17 it is evident that the gradient is higher where the base of the embankment is closer to the surface, i.e. at the toe, and very small where the base is far from the surface. This occurs because the temperature in the shallow portion of the subsoil decreases exponentially with depth. Unfortunately it has a drawback on piping detection that is worthwhile investigating. In the experiment described in this paper the dike rested directly on a sand layer prone to piping, so that the interface where piping developed (and where the temperature was monitored) was very shallow in the proximity of the toe. However, the piping-prone layer is often not located directly under the dike but is overlain by an impervious soil layer; it is therefore farther from the surface and the gradient along the seepage path could be not large enough to enable piping detection. In Fig. 18 the temperature difference between two points along the seepage path is represented as a function of depth. The aim is to represent the potential temperature difference between the head of a pipe and the toe of the embankment (where the sensor is located) for three different values of pipe length: 1, 3 and 5 m. It is assumed that a minimum temperature difference of 0.2 °C is necessary to guarantee piping detection. and the areas on the graphs where the temperature difference is lower than 0.2 °C are shaded. It turns out that if piping occurs at a depth smaller than 1.25 m from the ground level the initial gradient is large enough to enable piping detection the whole year. If piping occurs at a depth smaller than 5 m and the pipe is longer than 1 m the initial gradient is large enough to enable piping detection for most of the year.

It must be specified that the above simulations aim at providing a general framework rather than the exact temperature distribution and gradients under an embankment. As a matter of fact daily temperature variations and solar radiation were not taken into account in the model; very close to the toe (< 2 m) they can contribute to increase or decrease the gradient depending on the period of the year. Moreover the temperature distribution before strong seepage develops is depicted here; in real cases the hydraulic load increases gradually and the pipes could start developing when the initial gradient has already been altered to some degree by seepage.

The exploitation of the so-called initial gradient for piping detection has some affinity with the methods used to estimate exchanges between river or lakes and groundwater (see Sec. 1.4). The main limitation is that exploitation of seasonal temperature variations is not suitable for detecting seepage flows that have limited duration. Exploitation of daily temperature variations is suitable instead but possible only if the sensor is located at a very small depth (circa < 0.6 m).

## 6 Conclusions

The paper examines the effect of seepage and backward erosion piping on the temperature distribution under a dike, with specific attention to transient hydraulic conditions typical of river embankments.

Data from a field trial conducted on a purpose-built dike are presented that prove that a DTS sensor located circa 1 m behind the landside toe could detect temperature anomalies induced by backward eroding cavities in their early stage.

Thanks to the large amount of sensors installed, the experimental data also offer an overview of the temperature distribution under a dike, with and without seepage. The first thing deduced is that since soils prone to backward erosion piping are rather permeable heat transport by advection can occur, which means that the water entering at the riverside can reach the landside maintaining its initial temperature. Two parameters, the geothermal Péclet number and the thermal front, are suggested in this paper to ascertain, case by case, if a thermal front will propagate along the seepage path. Using these tools it is possible to predict if an advective front can travel from the riverside up to the head of a pipe before the critical water level is reached in the river, thus enabling its early detection. The geothermal Péclet number predicts the importance of advection over conduction in hydrogeological systems. Provided that advection is dominant, the thermal front velocity can be used to calculate the distance that an advective thermal front travels along the seepage path during a flood event. Approximately, for sands of high permeability advection prevails - or is as strong as conduction - for every possible size of the dike if the sand layer is thicker than 1. The time required for the advective front to reach the landside toe is in the order of a few days. For a medium permeability sand advection occurs at some degree (intermediate behaviour) if the sand layer is thicker than 3-4 m. However, the main constraint seems to be the time required by the front to reach downstream, which is in the order of tens of days for a medium/large dike.

Another possibility of identifying piping is given by the non-uniform temperature distribution at the base of a dike before seepage occurs. The onset of seepage produces an alteration of this initial temperature distribution. If advection prevails, as in the field trial, the temperatures vary both in the foundation soil and in the pipes, but at a faster rate in the latter. If conduction prevails the temperature varies only in the pipes, where the seepage rate is very high. A sensor located at the downstream toe of a dike can record these temperature variations immediately after seepage begins; detection of piping is thus enabled from the very beginning of the flood event. Numerical simulations suggest that when the permeable layer is located at a depth of more than 1 m from ground level the initial gradient might be not large enough to enable piping detection in some seasons.

One fibre deployed at the landside toe of a dike can be sufficient for piping detection, especially if the soil profile is very regular, but with more fibres deployed at different depths the reliability of the monitoring system is higher.

Considering the approximations made and the large number of variables at stake, future research involving numerical modelling of the temperature variations induced by piping and in-depth analysis of additional field data would be beneficial for the validation and further development of the concept developed in this work.

*Acknowlwdgements.* We are thankful to the Fondazione Cassa di Risparmio di Padova e Rovigo that founded the project Riversafe, on the use of fiber optic sensors for levee monitoring. The analysis of the field data was performed within the project. We also acknowledge TenCate Geosynthetics, eDF and geophyConsult for performing fibre optic measurements and providing their know-how about this technique.

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

**Table 1: Typical values of soil thermal properties.**

| Parameter | Value | Unit |
|---|---|---|
| Thermal conductivity of saturated sand | 2.8 | $\text{W m}^{-1}\,\text{K}^{-1}$ |
| Thermal conductivity of saturated clay | 1.4 | $\text{W m}^{-1}\,\text{K}^{-1}$ |
| Thermal conductivity of dry clay | 0.5 | $\text{W m}^{-1}\,\text{K}^{-1}$ |
| Specific heat capacity of water | 4186 | $\text{J kg}^{-1}\,\text{K}^{-1}$ |
| Volumetric heat capacity of sat. sand | $2.8 \cdot 10^{6}$ | $\text{J m}^{-3}\,\text{K}^{-1}$ |
| Volumetric heat capacity of sat. clay | $2.5 \cdot 10^{6}$ | $\text{J m}^{-3}\,\text{K}^{-1}$ |
| Volumetric heat capacity of dry clay | $1.6 \cdot 10^{6}$ | $\text{J m}^{-3}\,\text{K}^{-1}$ |

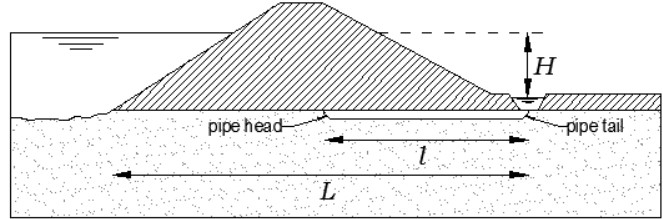

**Figure 1:  Backward erosion piping under a dike (adapted from ICOLD, 2015).**

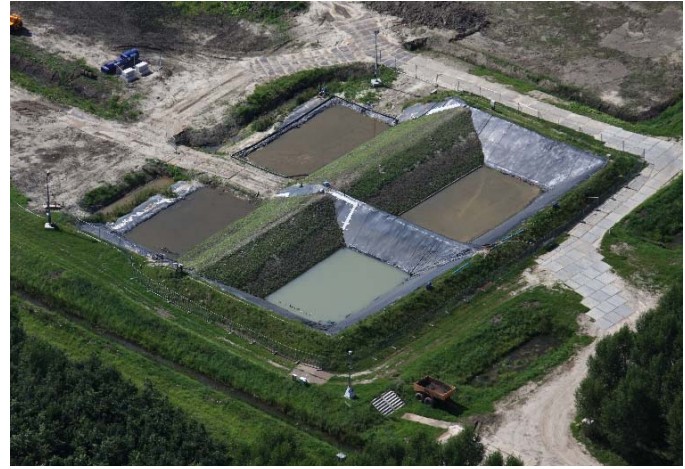

**Figure 2:  Large-scale piping facility**

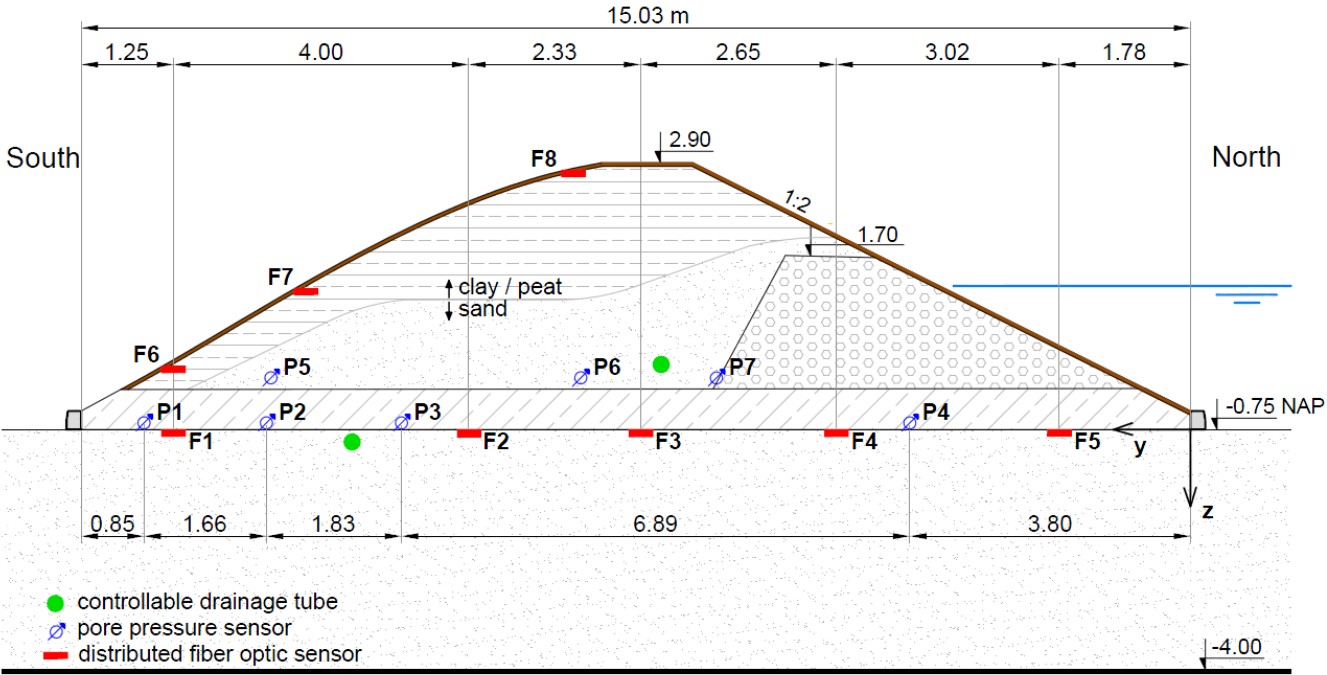

**Figure 3: Cross section of the test dike indicating the position of the sensors.**

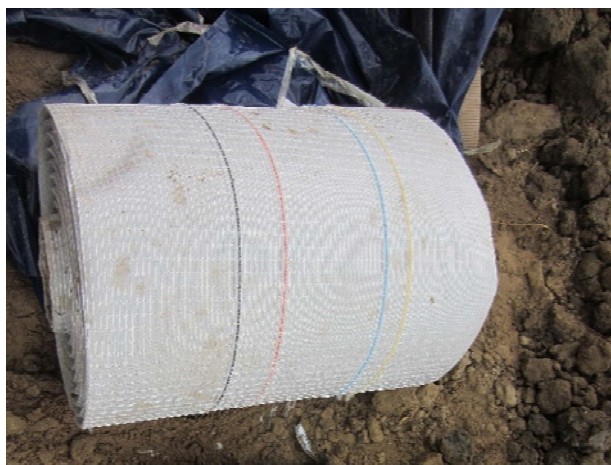

**Figure 4: Geotextile strip encasing two single-mode and two multi-mode optical fibres (courtesy of TenCate).**

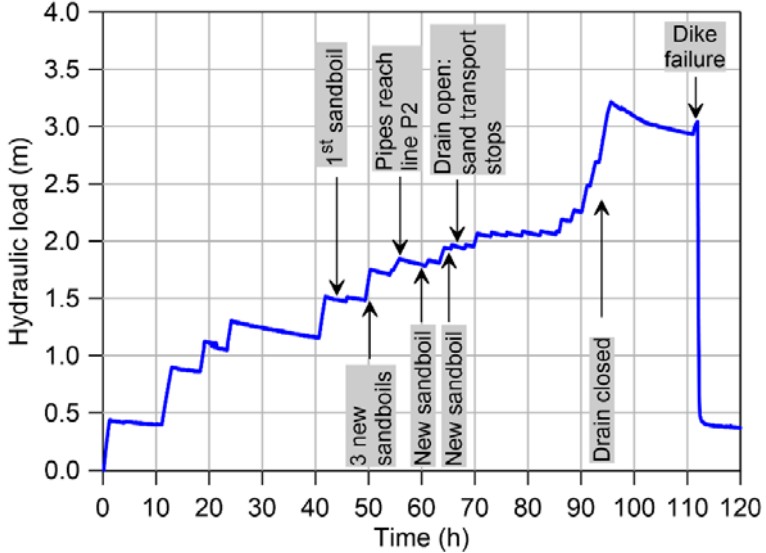

**Figure 5: Hydraulic load applied to the test dike.**

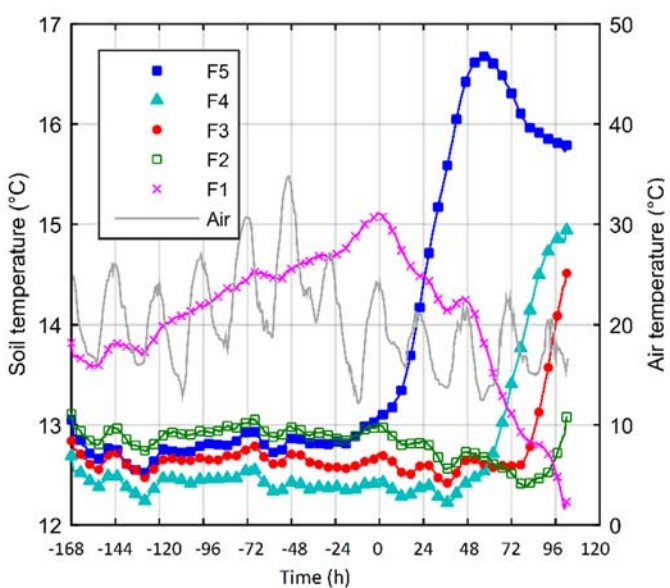

**Figure 6: Temperature measured by the distributed temperature sensor at different distances from upstream (F1 to F5) at x=10 m.**

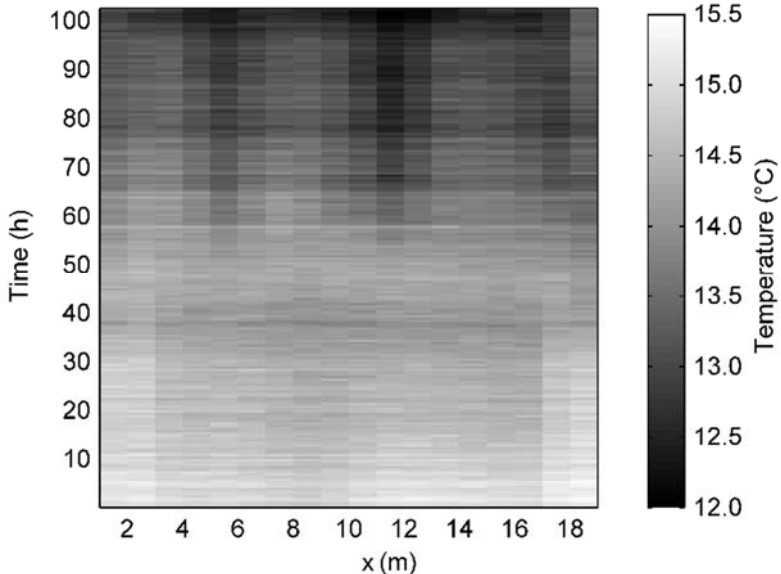

**Figure 7: Temperature measured along the dike near the downstream toe (F1).**

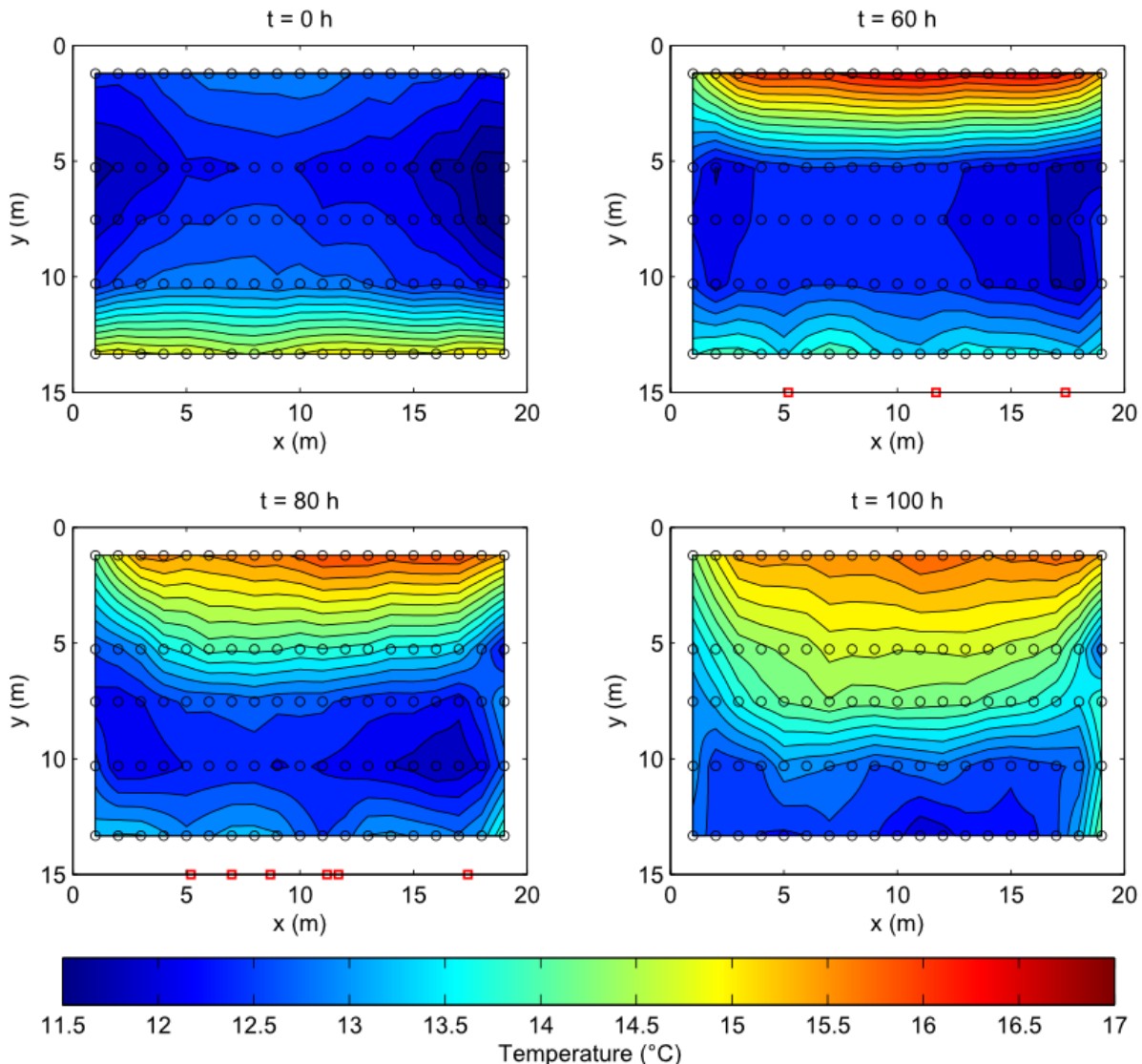

**Figure 8: Temperature at the bottom of the dike at 0, 60, 80 and 100 h from the beginning of the test (red squares indicate the position of sandboils and black circles the location of temperature sampling points).**

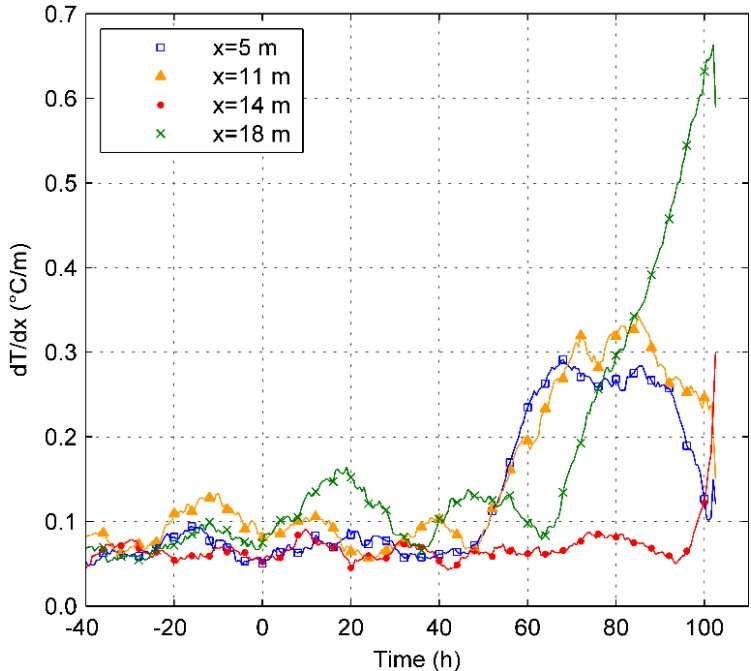

**Figure 9: Temperature gradients along the most downstream line F1 at some significant points: at x=5 m, x=11 m and x=18 m sand-boils were observed; at x=18 m also likely preferential seepage path at the contact with the foil; at x=14 m: no traces of piping (until 90 h when visual inspection was suspended).**

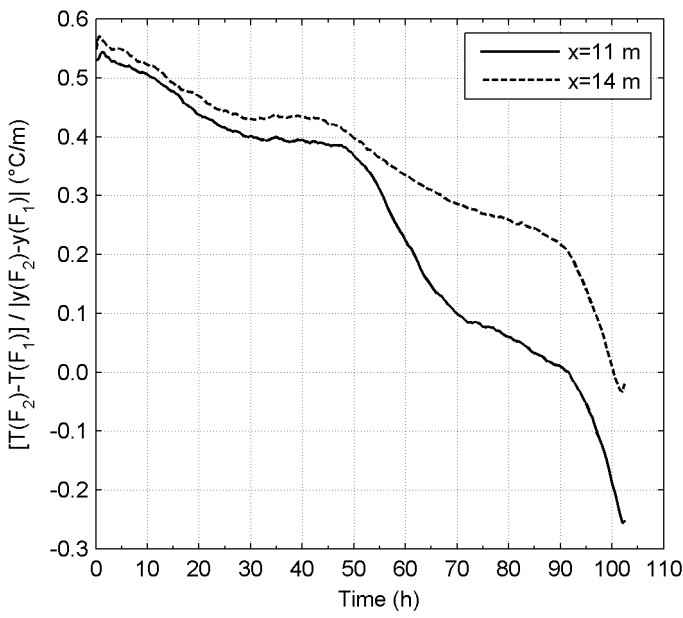

**Figure 10: Temperature gradients in the direction of the seepage flow at a location affected by piping (x=11 m) and at a location where no piping was detected (x=14 m).**

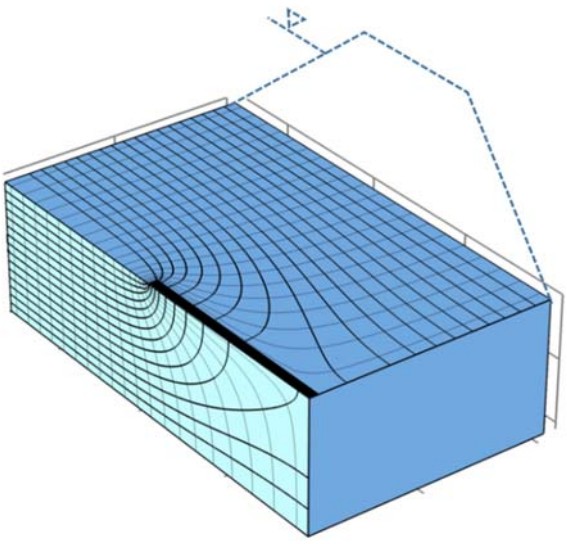

**Figure 11: Flow net around a pipe (black rectangle) extending for half of the seepage length. Exploiting symmetry, only half of the problem is modelled.**

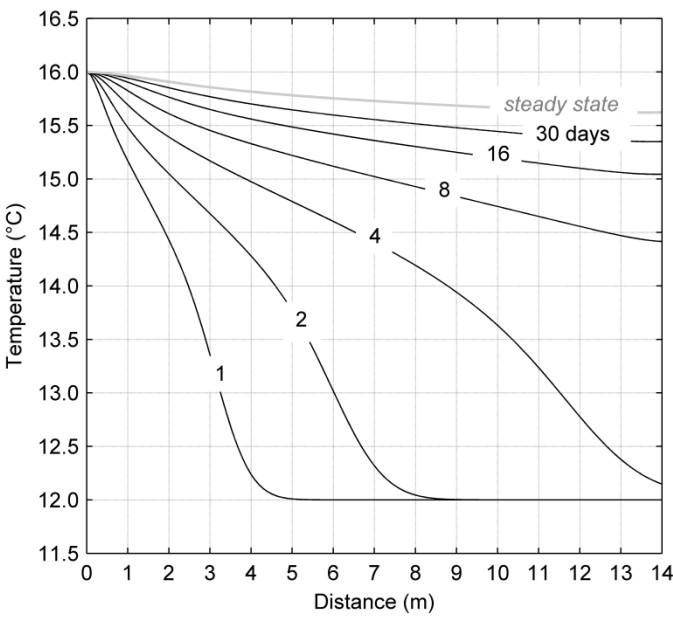

**Figure 12: FE simulation of a thermal front propagating in the foundation of the test dike for a constant hydraulic load of 3 m, constant inflow temperature of 16 °C and uniform initial temperature T=12 °C. On the x-axis the distance from the inflow point is indicated.**

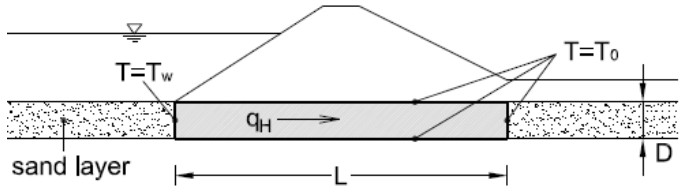

**Figure 13: Domain (grey rectangle) and boundary conditions assumed for the calculation of the geothermal Péclet number.**

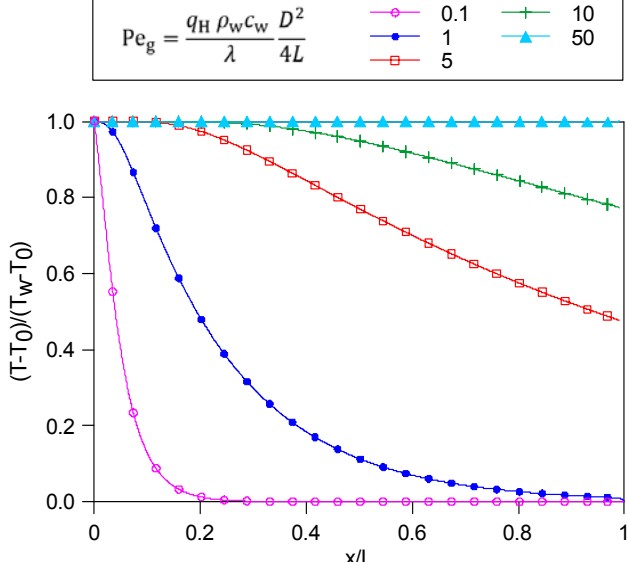

**Figure 14: Temperature distribution along the seepage path as a function of the geothermal Péclet number for the sand layer in Fig. 12. Results of a finite element parametric study in steady-state conditions.**

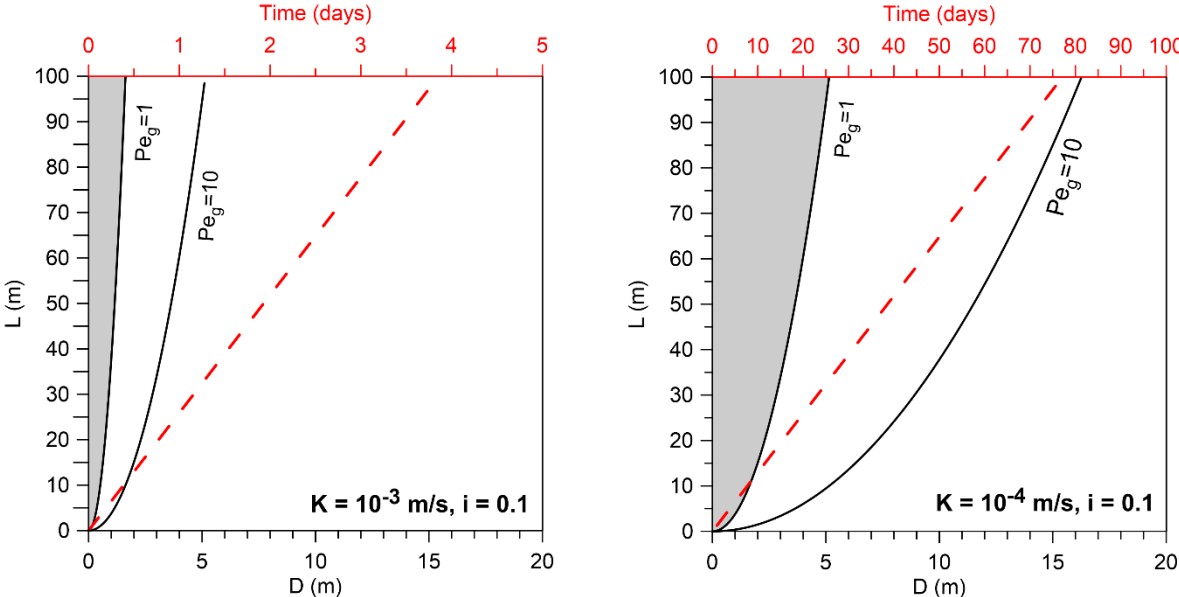

**Figure 15: Threshold values of the geothermal Péclet number as a function of distance along the seepage path (L) and thickness of the sand layer (D) for two different values of hydraulic conductivity. The dotted red line indicates the arrival times of the thermal front; these must be read on the top x-axis as a function of distance L.**

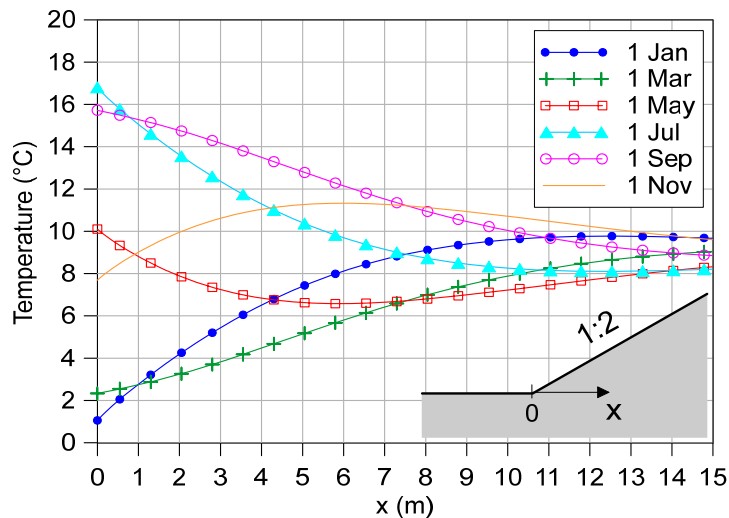

**Figure 16: Temperature at the base of a 1:2 slope from finite element modelling.**

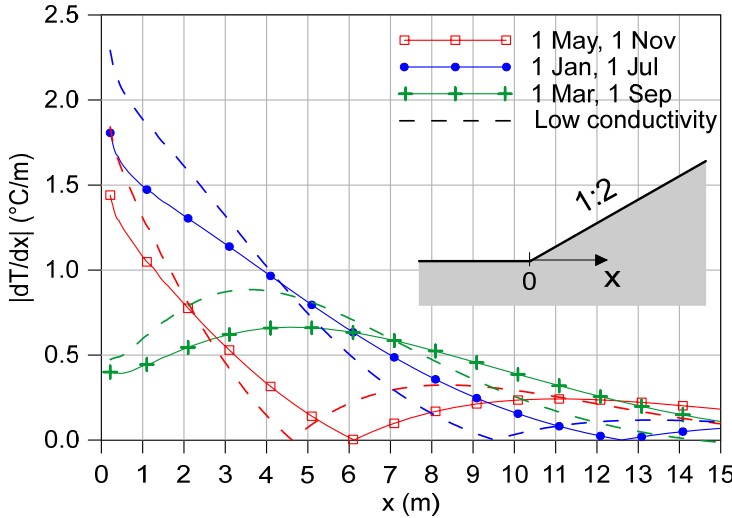

**Figure 17: Horizontal temperature gradient at the base of a 1:2 slope. Finite element solutions for average (solid line) and low (dashed lines) thermal conductivity of the embankment soil.**

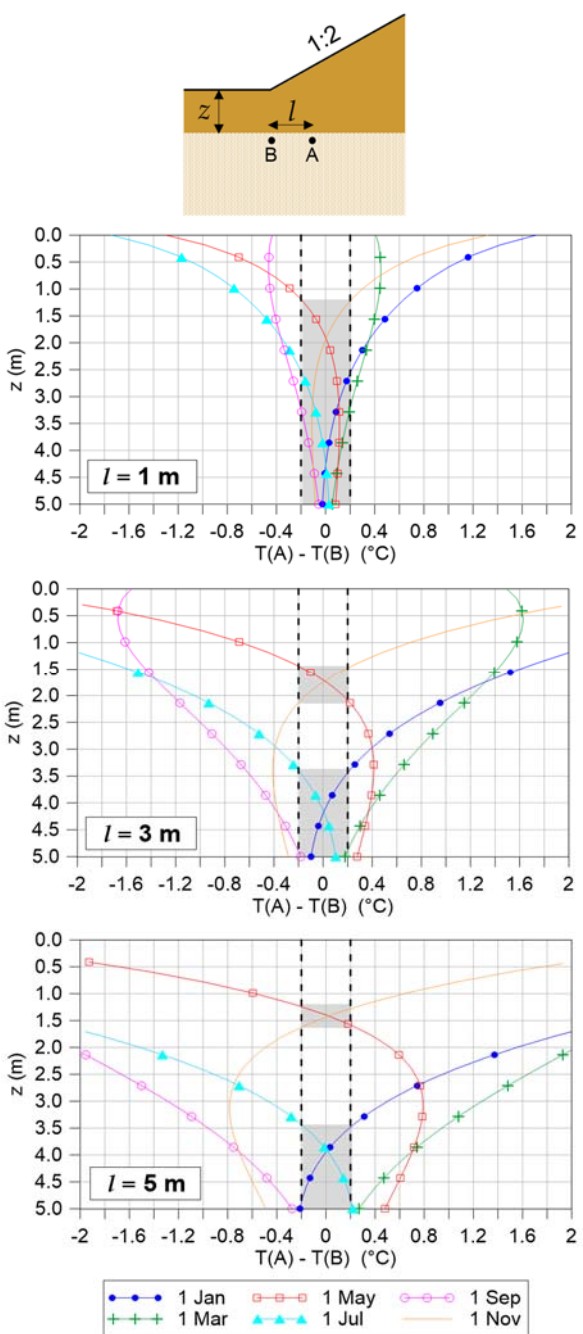

**Figure 18: Temperature difference between a point at the toe of the dike (B) and a point under the dike (A) as a function of depth from ground level (z). Point A is located at a distance *l* = 1, 3 and 5 m from the toe in the three graphs respectively. The regions of the graph where the temperature difference is smaller than 0.2 °C (in absolute value) are highlighted by grey shading.**