# Peer review of "Effectiveness of distributed temperature measurements for early detection of piping in river embankments"

_Hydrology and Earth System Sciences, 2016_

## Referee Comment (RC1)

**Effectiveness of distributed temperature measurements for early detection of piping in river embankments**
**Authors: Sylvia Bersan et al**

The manuscript discusses both the analysis of the results of a large-scale (so-called 'IJkdijk') experiment in the Netherlands to test the effectiveness of DTS in the early detection of backward piping erosion under dikes and a theoretical numerical study to predict thermal response below an embankment under seepage conditions. The study is of high relevance for the dike health monitoring community and extracts the relevant results of the IJkdijk experiments which at the moment have been described mainly in grey literature. It present novel data and the analysis to try to use DTS for piping erosion (internal erosion) is scientifically challenging. The paper has a strong geotechnical focus and terminology and somewhat less hydrological / earth scientific. Consequently the authors have not benefitted from the impressive amount of published research on DTS in hydrological systems. Moreover, I found the paper's structure confusing to say the least and the (modelling) assumptions not well explained and discussed. Maybe because it has been written like a work flow or learning curve of the authors. To tackle this, the paper needs major restructuring (I will elaborate on that below). In my opinion, the writing style and some figures also need some more work. Lastly, unfortunately, the paper seems to consist of two part (experiment and model) but the two parts do not get together. In conclusion, to me it is a potentially interesting paper but in current state it is not ready yet for publication in HESS.

In short I suggest restructuring as follows (details follow):
1 Intro
2 Theory and methods (including IJkdijk set up, maybe partly in supplement)
3 Perception and conceptual model of how DTS could detect piping
4 IJkdijk experiment results
5 Numerical results
6 Analysis/discussion of the both results and compare DTS and model
7 Conclude about possibility to detect piping and suggest optimal conditions and location

First, the authors start from DTS applications in geotechnics like dam monitoring but do not take advantage of the large amount of literature that is available in the hydrological community using high resolution DTS measurements to interpret the hydrological system, especially groundwater-surface water interaction (see work of e.g. Selker, Krause, Mamer, Fleckenstein, Cassidy, Briggs and many others to start a literature review). The here-described experiment has many similarities with published ground- surface water interaction studies and awkward statements can then be avoided like P3L27: "heat transfer in soils mostly occurs by conduction". So I suggest to write a generic intro on heat and water transport and how it can and will be used in your study. The paper should add comparison with published groundwater–surface water studies here and as such way stress the novelty and show the relevance for the hydrological community.

Second, the paper needs a clear methodology section in which the theory coupled heat and water flow modelling and the large-scale experiment are given. Describe both methods in the section 2 (bring 5.1 to section 2). How is the DTS calibrated? How did you make sure to distinct temperature from strain induced changes as there is quite some strain on the FO

cable which influences your measurements. The authors should clearly explain why the numerical model is needed/used (to what purpose do you model), why this formulation has been used and what assumption they made. Also add scientific references here. I think the main aspects of Bersan model (PhD thesis) should be included in the paper (maybe as supplement) as it is very relevant for this research. Furthermore, I found parts of this description unclear and confusing. For example, why you use specific discharge rate and not Darcian velocity, or P10L30 "neglects effect of conduction" followed by P10L31 "describes the effect of heat conduction". P11L8 the authors start writing annual temperature fluctuation under the assumption that "u=0" (P11L10). But this does not hold for your condition so elaborate on this assumption and why it can or in hindsight cannot be used. This effects strongly the interpretation of your results. Can the authors elaborate on this?

Third, the authors give some confusing if not conflicting description on what they expect from DTS under an embankment. Moreover, they do so in different parts of the paper. I suggest to include a structured section on the perceptional and conceptual model the authors have of using thermal measurements at the toe of the embankment (fig 13). This in terms of expected advective and conductive heat transport (see also first lines of section 4). A first statement on the expected flow travel time underneath a dike which you will elaborate later. Maybe also interesting to say something on expected temperature difference in river water during a year and during a flood (do you know if river water temperature changes a lot during typical flood events?). Basically, under which seepage conditions could DTS be informative for detecting piping (initiation) and when not. Obviously, if all water has identical temperature, nothing will be detected using DTS. Second, if the river water is warmer/colder than the GW, it is not expected to see piping developing at the toe of the embankment. In contrast to what the authors write (P7, L22-23), if you detect T differences due to river (reservoir) water the preferential flow path (possible pipe) already exists from river to toe of embankment and does not need to develop anymore. And your objective is detection the initiation and backward development of the pipe. Backward eroding cavities do not connect yet to the river and will logically first see inflow of the surrounding water (no T-difference). However, in case of warmer/colder water at the toe of an embankment compared to water underneath the embankment, piping initiation starting from the toe and eroding backwards, can be detected (as fortunately the case in this experiment). Note that strictly speaking one does not detect piping with DTS but rather preferential water flow. In my opinion such structured concepts will help describing your field data analysis and theoretical modelling results.

Fourth, the field results (current 3.3). This can be reduced to only data as concepts are given already. I suggest to the authors to stop describing the experiments after t=90hours as hereafter it is not a piping experiment using DTS anymore but a failure experiment which is not part of the paper's focus.

Lastly, the results of the dimensional and numerical analysis can be presented and the influence of the model and boundary assumptions be discussed. I, as a small example, do not understand why Figure 11 presents data using a constant hydraulic load of 3 m was used while the auhors have time series of hydraulic load during the experiment. The dimensional analysis presented in the manuscript allows to quantify the relative importance of conduction versus advection. This conclusion is important for the propagation of the front

assumption which requires advection to prevail. The results of such analysis are based on the assumptions of steady-state boundary conditions as acknowledged by the authors. Then the question arises if the model used for determining the thermal front propagation is representative of the actual conditions of the experimental set up?

This brings me to my final point, I was somewhat disappointed to see that the comparison of field and model results is quite minimal to almost absent. This is a missed opportunity and should be included in the paper.

Some minor points:
- Comparison with use of DTS in concrete dams seems not so relevant in your study
- Change terminology "Downstream and upstream of water under embankment" into "landward and river side"
- P4L10: "easily installed also in existing embankments" That is quite a statement. This only holds if the FO is installed in the toe (landward side) of the dike. But it is unclear if that is an optimal location for the FO in case of piping detection. I believe this is far from optimal location. Authors should explain this more.
- The derivation of the thermal front velocity starting from the heat transport equation in porous media is used to determine the "Retardation" coefficient. This coefficient is presented in equation 3 but its derivation is not yet clear.
- Time scale of figures could be homogenized to facilitate interpretation for reader
- Figure 6: It is written that the toe location temperature (F1) is highly influence by the air temperature whereas in the plot it is observed that it seems to have only influence between t= -7 days and t= -2 days air temperature is constant between t=0 days and t=5 days but the temperature at F1 continuously drops. Please explain.
- Why F1 reaches a temperature lower than any measurement in subsurface? Did you take into account that the FO cable could be strained due to displacement and therefore shows deviating temperature measurements (this links to the calibration question of the DTS.).
- Figure 7: plot this figure in color gradient for improving the readability.
- Figure 9 shows negative gradients in X=18. Why is this?
- Figure 10 could be explained based on the possible variability of the hydraulic conductivity.
- Figure 11 seems like is not considering the same initial conditions as the experiment. Also, the figure 8 at t=60h (approximately 2.5 days) shows a boundary temperature of around 14 degrees but the model results remain in 12 degrees. This may be interpreted as a large error taken into account that the steady state is achieved by a total increase of only 4 degrees.
- Figure 12 is not explained neither the variables or the system. This figure is a copy (very minor modification) of the original presented in Van der Kamp and Bachu 1989.
- Figure 15 and 16 show the temperature gradient along the x axis but it is not clear if they are located in y=0 or further. Please explain and compare with the experimental measurements.
- In addition, include the information of the moments when sand boils are observed as in figure 5 in other relevant figures.
- The grammar should be checked as there is an indiscriminate use of commas and frequent use of very long sentences.

---

## Referee Comment (RC2) · Anonymous Referee #2 · 13 Feb 2017

The article describes firstly a test carried out in the framework of the project IjkDijk to investigate an impact of the development of the piping in the foundation of a small-earth-damming-structure on the thermal-field of this structure and, secondly, it attempts to interpret the results of this test. Thirdly, it presents the results of numerical modelling that are not directly related to this test. The modelling was carried out to examine the temperature gradients that could be generated by internal erosion. The authors did not presented the relevant literature background related to the impact of the internal erosion including particularly the piping influence on thermal field of soil/damming structures. The conclusions from literature could be helpful in the interpretation by the authors of results of their test. Some of the presented conclusions are already

described in the literature. The chapters should have more logical sequence and arrangement. In my opinion this is interesting paper in terms of the verification of the suitability of thermal method in application to piping detection. However, the article requires significant adjustments before publication in HESS.

**The order of chapters should be changed. Description of model and theory concerning the geothermal Péclet number should be presented before the trial test description.**

**The authors presented in the introduction some information about previous research concerning mechanical aspect of piping process and failure criteria. However they did not mentioned the existing research results concerning main goal of the paper, i.e., about the thermal influence of internal erosion on soil/damming-structure temperature field, and particularly of the backward erosion and piping thermal influence. Relevant information can be found in a thesis of Guidoux (2008), a thesis of Radzicki (2009), two papers of Radzicki and Bonelli from 2010 and Bonelli and Radzicki (2012). Moreover, a similar test, like described in this paper, was carried out in 2009, also in the framework of the IjkDijk project that gave some interesting and important results. This test is not mentioned in the reviewed paper.**

**The title of chapter 2 "Early detection of internal erosion" should be changed. It does not correspond well to the content of this chapter.**

**The chapter 3.3 contains some information that could have been presented] in chapter 3.4, and conversely]. I suggest to merge them in one chapter.**

**The authors use often the words "erosion" or "internal erosion". It would be more clear if there be an explanation what the internal erosion is and that piping (backward erosion piping) is one of the internal erosion processes.**

**p.1, rows 9-11 "This work investigates the effectiveness of DTS for dike monitoring, focusing on early detection of backward erosion piping, a mechanism that affects the foundation layer of structures resting on permeable, sandy soils." Remark: Reader**

may think that the piping always affects the foundation layer only. I propose to write for example "...focusing on early detection of backward erosion piping for the case of a mechanism that affects the foundation layer..."

**p.3, row 8 "The use of DTS for early detection of internal erosion is nowadays common practice in dam monitoring" Remark : Bibliographic references to confirm this conclusion should be included**

**p.3, rows 23-27 "Among these are electrical resistivity, self‐potential and temperature (Sheffer et al., 2009). These methods have a common feature: they provide spatially distributed measurements. Unfortunately they also have a common shortcoming: they measure quantities that are influenced by a large number of variables besides the occurrence of internal erosion, which makes data interpretation not straightforward and ambiguous." Remark: Indeed, the electrical resistivity and self‐potential measure quantities that are particularly influenced by a large number of variables. Contrary, the thermal method is influenced mostly by humidity variation and water flow.**

**p.3, row 30 "Because of conduction...." Remark: It would be more precisely to write "In the case of only conduction heat transport in the soil (without water flow) ..."**

**p.4, rows 1-2 "While moderate seepage flow occurring in the soil does not affect the temperature field determined by heat conduction, significant seepage (rates higher than 10-7 - 10-6 m/s) produce variations in the soil temperature." Remark: Temperature distribution in soil, including changes of temperature field generated by seepage depend not only on water velocity but also on length of the seepage path, so we can say as well that temperature distribution depends on Péclet number that contains both these variables. The same water velocity results in different temperature field variation depending on the scale of a damming structure. In consequence, the authors should better explain their conclusion, referring it for example only to a limited range of the scale of damming structures.**

**p.4, rows 4-6 "Johansson and Sjödal (2009) observed that, in dams, the regions af-**

[Figure]

fected by internal erosion are typically characterized by a temperature similar to the reservoir water, while the temperature in the other regions of the flow domain is nearly unaffected by the reservoir temperature." Remark: It is not clear what the authors explain by this paragraph. If it is an example to confirm the previous sentence that "Since internal erosion promotes the formation of zones of higher permeability and, consequently, a local increase of seepage rate, it is expected that it also causes a local variation of the temperature field "they should add "for example Johansson and Sjö-dal (2009) observed that, …." There are different possible results of internal erosion development on temperature field. Research results of Radzicki and Bonelli (2012), Radzicki and Bonelli (two papers in 2010) explained that the thermal influence of internal erosion depends on: - the permeability of the soil which surrounds the zone of internal erosion - the type of internal erosion process and level of its development (geometrical and/or mechanical) - the scale of a damming structure For example, one of possible results of internal erosion development is that the water reservoir temperature is transported only to the upstream side of the damming structure. In consequence, on one hand there is no effect of water reservoir temperature on the downstream side of damming structure temperatures. On the other hand, the transport of heat from the downstream slope into the body of the damming structure is affected by local acceleration of water flow due to the internal erosion development, which allows to detect leakages and internal erosion.

**p.8, rows 22-26 "For this reason, the mechanism that promoted the formation of piping-induced thermal anomalies was not, as commonly found in the literature, the prevalence of advection in the eroded regions in opposition to the purely conductive behaviour of the regions unaffected by piping. This occurs because the assumption of conductive behaviour in the unaffected regions (Johansson and Hellström 2001; Johansson and Sjödal 2009) only applies to low permeability bodies such as dam cores.." Remark : The thermal effect of piping development in soil, comparing with the case of soil without piping, was analyzed for different value permeability of soil and for different advection transport intensity (Péclet value from 0.1 up to 100) in the thesis of**

Radzicki (2009) and presented as well in Radzicki and Bonelli (2010).

Bibliography: # Guidoux C. (2008). Développement et validation d'un système de détection et de localisation par fibres optiques de zones de fuite dans les digues en terre. PhD rapport. # Radzicki K. (2009), Analyse retard des mesures de températures dans les digues avec application à la détection de fuites. PhD rapport. # Radzicki K., Bonelli S. (2010), A possibility to identify piping erosion in earth hydraulic works using thermal monitoring, 8h ICOLD European Club Symposium, p. 618-623. # Radzicki K. Bonelli S. (2010) Thermical seepage monitoring in the earth dams with Impulse Response Function Analysis model, 8h ICOLD European Club Symposium, p. 649-654. # Radzicki K., Bonelli S. (2012), Monitoring of the suffusion process development using thermal analysis performed with IRFTA model. 6th ICSE, p.593-600.

---

## Author Comment (AC1) · 4 Apr 2017

**Effectiveness of distributed temperature measurements for early detection of piping in river embankments**

**REPLY TO COMMENTS BY REVIEWER 1**

*Corresponding author: Silvia Bersan*

The comments from the reviewer are very useful and will certainly help improving the paper.

The main requests by Reviewer 1 are:

- add references and comparison to previous works in the field of surface water - groundwater interaction;
- rearrange the structure of the paper: group the theory in a single chapter (also asked by reviewer 2) and add a section with a conceptual model about the functioning of DTS for piping detection;
- link experimental data and numerical model and clarify some aspect of the numerical model.

I went through the suggested literature and I agree that it is relevant to this study. I rearranged the paper as suggested and I found that the new structure is much clearer. Finally, I acknowledge that some details regarding the numerical models and a better comparison between numerical and experimental data would make the paper more clear.

In the following I will answer to the comments (reported in italics) point by point, also on behalf of the co-authors.

*First, the authors start from DTS applications in geotechnics like dam monitoring but do not take advantage of the large amount of literature that is available in the hydrological community using high resolution DTS measurements to interpret the hydrological system, especially groundwater-surface water interaction (see work of e.g. Selker, Krause, Mamer, Fleckenstein, Cassidy, Briggs and many others to start a literature review). The here-described experiment has many similarities with published ground- surface water interaction studies and awkward statements can then be avoided like P3L27: "heat transfer in soils mostly occurs by conduction". So I suggest to write a generic intro on heat and water transport and how it can and will be used in your study. The paper should add comparison with published groundwater–surface water studies here and as such way stress the novelty and show the relevance for the hydrological community.*

I will briefly describe in the introduction the working principle of the thermometric method and some data interpretation methods used in dam monitoring and in groundwater–surface water studies. I will try to add comparisons in the discussion of the data.

I will also try to explain what is the novelty of the study. In my opinion the main novelty consists in the fact that data interpretation models so far mostly consider steady water flow. This work focuses on the effect of a transient flow of small duration. The work also focuses on permeable soils, which are the soils most prone to backward erosion piping, in opposition to low permeability soils of which dams are made of. The effect of the size of the water retaining structure is also investigated in this paper.

*Awkward statements can then be avoided like P3L27: "heat transfer in soils mostly occurs by conduction"*

That can be rearranged as:

"The [thermometric] method is based on the principle that, in the absence of seepage (or in presence of moderate seepage flow), the temperature distribution in the upper portion of the subsoil fluctuates seasonally in response to the variations of the air temperature."

*Second, the paper needs a clear methodology section in which the theory coupled heat and water flow modelling and the large-scale experiment are given. Describe both methods in the section 2 (bring 5.1 to section 2).*

I do agree. Section 5.1 and the first part of 5.2 have been moved to section 2

*How is the DTS calibrated?*

The measurements were performed by a company different form the company that designed and led the test. The calibration procedure was not specified. It was guaranteed that temperature changes were measured with an accuracy of 0.1°C.*
* O. Artières and G. Dortland (2012) IJkdijk All In One Sensor Validation Test. Report of the measurements made with TenCate GeoDetect® system on the East, West and South dikes, TenCate Geosynthetics

*How did you make sure to distinct temperature from strain induced changes as there is quite some strain on the FOS cable which influences your measurements?*

The DTS system exploited the Raman effect, which is not sensitive to strain.

*The authors should clearly explain why the numerical model is needed/used (to what purpose do you model), why this formulation has been used and what assumption they made. Also add scientific references here.*
*I think the main aspects of Bersan model (PhD thesis) should be included in the paper (maybe as supplement) as it is very relevant for this research. Furthermore, I found parts of this description unclear and confusing.*

The model is used to understand the impact that factors as geometry, time-scale of the hydraulic loads, initial conditions, thermal dispersion etc. have on the temperature field. The results in Fig. 11 show the effect of time for a given geometry (the geometry of the test dike)
Dimensional analysis is used to clarify the effect of geometry for steady state conditions. A (steady state) numerical model is used to verify the soundness of the dimensional analysis. This model is not intended to reproduce the field measurements but to extrapolate the results to a more general context.
The path we chose to generalize the results of the test was splitting the problem in two parts: a steady state part, addressed using dimensional analysis, and a transient part, addressed using the thermal front velocity.
I will include more details about the numerical model.

*For example, why you use specific discharge rate and not Darcian velocity*

With specific discharge rate I mean Darcian velocity. I have now clarified it in the theory section.

*or P10L30 "neglects effect of conduction" followed by P10L31 "describes the effect of heat conduction". P11L8 the authors start writing annual temperature fluctuation under the assumption that "u=0" (P11L10). But this does not hold for your condition so elaborate on this assumption and why it can or cannot be used. This effects strongly the interpretation of your results. Can the authors elaborate on this?*

Good point. I will clarify this in the paper.

The assumption $u$=0 holds true if we want to determine the temperature of the test embankment at the beginning of the test, because no seepage occurred from the moment the embankment was built until the beginning of the test.

The results are also valid for real cases where $u$ is small enough that advection is negligible (excluding sporadic flood events). This holds true unless we are dealing with a strongly gaining or loosing stream. Whether advection is negligible depends on the hydraulic gradient between river and groundwater and the hydraulic conductivity of the sediments.

Therefore, the condition is not always satisfied in the field but I believe it is representative of a category of a large number of real cases.

*Third, the authors give some confusing if not conflicting description on what they expect from DTS under an embankment. Moreover, they do so in different parts of the paper. I suggest to include a structured section on the perceptional and conceptual model the authors have of using thermal measurements at the toe of the embankment (fig 13). This in terms of expected advective and conductive heat transport (see also first lines of section 4). A first statement on the expected flow travel time underneath a dike which you will elaborate later.*

I do agree on the fact that a section containing a conceptual model would be useful. So far, I prefer to insert this section after that presentation of the data from the field test, because I consider the conceptual model an output of the research, since it was elaborated observing the data. The data will be then presented with no or little interpretation in section 3 and will be discussed in the section 4 where the conceptual model will be elaborated.

*Maybe also interesting to say something on expected temperature difference in river water during a year and during a flood (do you know if river water temperature changes a lot during typical flood events?). Basically, under which seepage conditions could DTS be informative for detecting piping (initiation) and when not. Obviously, if all water has identical temperature, nothing will be detected using DTS. Second, if the river water is warmer/colder than the GW, it is not expected to see piping developing at the toe of the embankment. In contrast to what the authors write (P7, L22-23), if you detect T differences due to river (reservoir) water the preferential flow path (possible pipe) already exists from river to toe of embankment and does not need to develop anymore. And your objective is detection the initiation and backward development of the pipe.*

A preferential flow path already existing between river and landside could represent a region of higher permeability (f.i. a paleochannel), which is the best location for the formation of a pipe. Therefore this information is useful for identifying spots where the risk of piping is higher rather than for detecting existing pipes. I will explain this in the text.

*Backward eroding cavities do not connect yet to the river and will logically first see inflow of the surrounding water (no T-difference). However, in case of warmer/colder water at the toe of an embankment compared to water underneath the embankment, piping initiation starting from the toe and eroding backwards, can be detected (as fortunately the case in this experiment). Note that strictly speaking one does not detect piping with DTS but rather preferential water flow. In my opinion such structured concepts will help describing your field data analysis and theoretical modelling results.*

I'll try to make use of this description in rearranging the conceptual model.
*Fourth, the field results (current 3.3). This can be reduced to only data as concepts are given already.*

As said earlier, the data will be presented with little interpretation next to it in section 3 and will be further discussed in the section 4 where the conceptual model will be elaborated.

*I suggest to the authors to stop describing the experiments after t=90hours as hereafter it is not a piping experiment using DTS anymore but a failure experiment which is not part of the paper's focus.*

I think temperature data after 90 h are still interesting and helped in elaborating a point in the conceptual model that would not be otherwise possible to explain.

*Lastly, the results of the dimensional and numerical analysis can be presented and the influence of the model and boundary assumptions be discussed.*
*I, as a small example, do not understand why Figure 11 presents data using a constant hydraulic load of 3 m was used while the authors have time series of hydraulic load during the experiment.*
*The dimensional analysis presented in the manuscript allows to quantify the relative importance of conduction versus advection. This conclusion is important for the propagation of the front assumption which requires advection to prevail. The results of such analysis are based on the assumptions of steady-state boundary conditions as acknowledged by the authors. Then the question arises if the model used for determining the thermal front propagation is representative of the actual conditions of the experimental set up?*

First, the model was used to reproduce the experiment. The complete model had non-homogeneous initial conditions and variable inflow temperature as well as variable hydraulic gradient. Some details will be given about this model.
* Once calibrated, the model was used to run other analysis to extend the results to a number of field situations. Homogeneous initial temperature, constant temperature of the inflow water and constant hydraulic load were used as a more generic boundary conditions. Therefore data figure 11 are not to be compared directly with the results from the test. This will be clarified in the text.
Next to the model reproducing the experiment, another model with constant hydraulic load has been run. This was done to simplify the problem and highlight the time scale of the problem. This can be better explained in the text to avoid erroneous comparisons between model and data.
The model was always run in transient mode, initially applying transient BC representing the actual conditions at the experimental site and, later, constant BC condition to simplify the problem. This will be specified better in a brief description of the numerical model.

Minor points:

*- Comparison with use of DTS in concrete dams seems not so relevant in your study*

In some concrete dams the sensor is placed in the foundation soil to detect underseepage. In concrete face dams DTS is used to detect leakage from joints. The basic working principle is the same. I will clarify it.

*- Change terminology "Downstream and upstream of water under embankment" into "landward and river side"*

I agree. In the description of the field test I'll keep using upstream-downstream, but in the description of the conceptual model and other discussions I'll use riverside and landside, because when talking about rivers the terms upstream and downstream would be otherwise confusing.

*- P4L10: "easily installed also in existing embankments" That is quite a statement. This only holds if the FO is installed in the toe (landward side) of the dike. But it is unclear if that is an optimal location for the FO in case of piping detection. I believe this is far from optimal location. Authors should explain this more.*

True. This is maybe more accurate:

"Optical cables can now be installed both in new and existing embankments lengthwise the dike. In existing embankments the cable is generally placed at the toes or along the slopes, since installation at these locations simply require the excavation of a narrow trench. Installation at other locations would require the use of directional drilling or similar techniques."

I will also elaborate about the optimal position of the sensor in the discussion or conclusions.

*- The derivation of the thermal front velocity starting from the heat transport equation in porous media is used to determine the "Retardation" coefficient. This coefficient is presented in equation 3 but its derivation is not yet clear.*

I don't understand what the reviewer would like to find in the text. Showing the derivation of the retardation coefficient means showing how the advection-diffusion equation is determined for a porous medium, which is out of scope here. Maybe adding a reference to the theory is beneficial. For instance: J. Bear (1972) *Dynamics of Fluids in Porous Media*, American Elsevier Publishing Company, Inc., New York.

*- Time scale of figures could be homogenized to facilitate interpretation for reader*

I agree.

*- Figure 6: It is written that the toe location temperature (F1) is highly influence by the air temperature whereas in the plot it is observed that it seems to have only influence between t= -7 days and t= -2 days air temperature is constant between t=0 days and t=5 days but the temperature at F1 continuously drops. Please explain.*

The temperature at F1 is highly influenced by the ait temperature when there is no water flow, therefore before the beginning of the test (t=0). In presence of water flow the temperature at F1 is determined by the temperature of the seepage water.

*- Why F1 reaches a temperature lower than any measurement in subsurface? Did you take into account that the FO cable could be strained due to displacement and therefore shows deviating temperature measurements (this links to the calibration question of the DTS.).*

The lowest temperature measured at F1 is very likely the temperature at the base of the dike at the beginning of the test somewhere between line F3 and F4. It is likely that none of the 5 measuring lines was placed at the coldest spot under the dike. The coldest spot is not exactly under the crest because of the exposition of the dike with the slopes facing south and north.

- Figure 7: plot this figure in color gradient for improving the readability. OK

- Figure 9 shows negative gradients in X=18. Why is this? I don't see negative gradients in Figure 9.

*- Figure 10 could be explained based on the possible variability of the hydraulic conductivity.*

Although the test dike was built as homogeneous as possible, some minor heterogeneities were present due to replacements of materials after a previous series of tests.

However, in this case the gradients at the two locations start diverging exactly after the developments of the first pipes…quite suspicious.

*- Figure 11 seems like is not considering the same initial conditions as the experiment. Also, the figure 8 at t=60h (approximately 2.5 days) shows a boundary temperature of around 14 degrees but the model results remain in 12 degrees. This may be interpreted as a large error taken into account that the steady state is achieved by a total increase of only 4 degrees.*

See answer above (*)

*- Figure 12 is not explained neither the variables or the system. This figure is a copy (very minor modification) of the original presented in Van der Kamp and Bachu 1989.*

The figure is indeed not very informative. I will eliminate it.

*- Figure 15 and 16 show the temperature gradient along the x axis but it is not clear if they are located in y=0 or further. Please explain and compare with the experimental measurements.*

The coordinate x in Fig. 15 and 16 is the coordinate y in the graphs depicting the results of the field test. I could change it if it is confusing. The measurements of field test can only be qualitatively compared to these results, because the test embankment was built only a couple of months before the test. For comparison, we can only consider the north slope where the influence of solar radiation can be neglected. The temperature variation between line F4 and F5 is 1°C while the model predicts 0.5°C.

*- In addition, include the information of the moments when sand boils are observed as in figure 5 in other relevant figures.*

Ok. I would do that for figure 7, 9 and 10.

*- The grammar should be checked as there is an indiscriminate use of commas and frequent use of very long sentences.*

---

## Author Comment (AC2) · 4 Apr 2017

**Effectiveness of distributed temperature measurements for early detection of piping in river embankments**

**REPLY TO COMMENTS BY REVIEWER 2**

*Corresponding author: Silvia Bersan*

I thank the reviewer for the suggestions that I found very useful. In the following I answer all his comments also on behalf of the co-authors.

*# The order of chapters should be changed. Description of model and theory concerning the geothermal Péclet number should be presented before the trial test description.*

*# The title of chapter 2 "Early detection of internal erosion" should be changed. It does not correspond well to the content of this chapter.*

*# p.3, row 8 "The use of DTS for early detection of internal erosion is nowadays common practice in dam monitoring" Remark: Bibliographic references to confirm this conclusion should be included*

*# The authors presented in the introduction some information about previous research concerning mechanical aspect of piping process and failure criteria. However they did not mentioned the existing research results concerning main goal of the paper, i.e., about the thermal influence of internal erosion on soil/damming-structure temperature field, and particularly of the backward erosion and piping thermal influence. Relevant information can be found in a thesis of Guidoux (2008), a thesis of Radzicki (2009), two papers of Radzicki and Bonelli from 2010 and Bonelli and Radzicki (2012). Moreover, a similar test, like described in this paper, was carried out in 2009, also in the framework of the IJkdijk project that gave some interesting and important results. This test is not mentioned in the reviewed paper.*

The structure of the paper has been changed, also upon suggestion of the other reviewer. Previous section 5.1 and part of 5.2, introducing the Peclét number, are now in section 2. Section 2 contains theory and methods, the latter consisting in the description of the field test.

The content of previous chapter 2 "Early detection of internal erosion" has been moved to the Introduction chapter in the section 1.3 "the thermometric method". This section will also include references that confirm that "The use of DTS for early detection of internal erosion is nowadays common practice in dam monitoring" and previous research about data interpretation methods and thermal influence of internal erosion on soil/damming-structure temperature field.

Comparison with the 2009 IKdijk experiment, which dynamic was a bit different, could be made in the discussion of the data.

*# The chapter 3.3 contains some information that could have been presented] in chapter 3.4, and conversely]. I suggest to merge them in one chapter.*

I agree it was confusing. I limited 3.3 to the presentation and basic interpretation of the data. I moved the interpretation in previous section 3.4 to a new section 4 containing a conceptual model of detection of piping by means of temperature measurements.

*# The authors use often the words "erosion" or "internal erosion". It would be more clear if there be an explanation what the internal erosion is and that piping (backward erosion piping) is one of the internal erosion processes.*

I added:
"Backward erosion piping is a specific kind of internal erosion. Internal erosion is the removal and transport of soil particles operated by seepage flow within an embankment or its foundation."
And a foot note:
"For the sake of brevity the term *piping* is used in the paper to indicate backward erosion piping. In the literature the term piping is also used as a synonym of internal erosion or to indicate other types of internal erosion, as concentrated erosion (f.i. the formation of a pipe through the core of a dam) or contact erosion (see ICOLD, 2015)"

*# p.1, rows 9-11 "This work investigates the effectiveness of DTS for dike monitoring, focusing on early detection of backward erosion piping, a mechanism that affects the foundation layer of structures resting on permeable, sandy soils." Remark: Reader may think that the piping always affects the foundation layer only. I propose to write for example "...focusing on early detection of backward erosion piping for the case of a mechanism that affects the foundation layer: : :"*

*# p.3, row 8 "The use of DTS for early detection of internal erosion is nowadays common practice in dam monitoring" Remark : Bibliographic references to confirm this conclusion should be included*

I added: "Backward erosion piping consists in the formation of a thin pipe below a roof provided by a layer of cohesive soil or a rigid structure. It mostly occurs in the foundation of water retaining structures when the foundation soil consists of uniform, fine to medium sand (ICOLD, 2015). [...] Backward erosion piping can also occur, although less frequently, under revetments (Galiana, 2005) or in riverbanks and dike bodies with sandy inclusions (Hagerty, 1991)."

*# p.3, rows 23-27 "Among these are electrical resistivity, self-potential and temperature (Sheffer et al., 2009). These methods have a common feature: they provide spatially distributed measurements. Unfortunately they also have a common shortcoming: they measure quantities that are influenced by a large number of variables besides the occurrence of internal erosion, which makes data interpretation not straightforward and ambiguous." Remark: Indeed, the electrical resistivity and self- potential measure quantities that are particularly influenced by a large number of variables. Contrary, the thermal method is influenced mostly by humidity variation and water flow.*

For transient water flow and shallow measurements (i.e. measurements at the toe of an embankment or inside an embankment lower than 10 m) also the initial temperature distribution in/under the embankment influences the thermal response of the structure. Solar radiation, rain (see the work by A. Khan), wind also influence the temperature field in the shallow portion of the subsurface. So I believe the problem is rather complex, but, as you stated, less that complex than the inversion of resistivity and self-potential fields, see f.i. Rittgers et al. (2015)* for the analysis of a similar experiment at the IJkdijk test site.

By relating temperature measurements along the dam toe with measurements in the transverse direction (quasi 3D systems) I believe that some ambiguities could be resolved.

*J.B. Rittgers, A. Revil, T. Planes, M.A. Mooney and A.R. Koelewijn (2015) "4-D imaging of seepage in earthen embankments with time-lapse inversion of self-potential data constrained by acoustic emissions localization", *Geophys. J. Int*., **200**, 758–772.

*# p.3, row 30 "Because of conduction...." Remark: It would be more precisely to write "In the case of only conduction heat transport in the soil (without water flow) ..."*

It was rearranged as: "In the absence of seepage (or in presence of moderate seepage flow), the temperature distribution in the upper portion of the subsoil fluctuates seasonally in response to the variations of the air temperature".

*# p.4, rows 1-2 "While moderate seepage flow occurring in the soil does not affect the temperature field determined by heat conduction, significant seepage (rates higher than 10-7 - 10-6 m/s) produce variations in the soil temperature." Remark: Temperature distribution in soil, including changes of temperature field generated by seepage depend not only on water velocity but also on length of the seepage path, so we can say as well that temperature distribution depends on Péclet number that contains both these variables. The same water velocity results in different temperature field variation depending on the scale of a damming structure. In consequence, the authors should better explain their conclusion, referring it for example only to a limited range of the scale of damming structures.*

This is correct, thanks for the remark. Since the basics of the thermometric methods are explained before introducing the Péclet number, I preferred to removed any value of Darcy velocity in that part.
Immediately after introducing the Peclet number I wrote: "Assuming a characteristic length of 10 m and typical values of the thermal properties of soils (see Table 1) the Péclet number is on the order of 1 when the Darcy velocity is on the order of 10-7 m/s"
The influence of the geometry is then further explained. In our theoretical model both the thickness of the sand layer and the size of the damming structure play a role.

*# p.4, rows 4-6 "Johansson and Sjödal (2009) observed that, in dams, the regions affected by internal erosion are typically characterized by a temperature similar to the reservoir water, while the temperature in the other regions of the flow domain is nearly unaffected by the reservoir temperature." Remark: It is not clear what the authors explain by this paragraph. If it is an example to confirm the previous sentence that "Since internal erosion promotes the formation of zones of higher permeability and, consequently, a local increase of seepage rate, it is expected that it also causes a local variation of the temperature field "*

I agree on the fact that this sentence in that position was confusing. I will improve it.

*they should add "for example Johansson and Sjödal (2009) observed that, : : :." There are different possible results of internal erosion development on temperature field. Research results of Radzicki and Bonelli (2012), Radzicki and Bonelli (two papers in 2010) explained that the thermal influence of internal erosion depends on: - the permeability of the soil which surrounds the zone of internal erosion - the type of internal erosion process and level of its development (geometrical and/or mechanical) - the scale of a damming structure For example, one of possible results of internal erosion development is that the water reservoir*

*temperature is transported only to the upstream side of the damming structure. In consequence, on one hand there is no effect of water reservoir temperature on the downstream side of damming structure temperatures. On the other hand, the transport of heat from the downstream slope into the body of the damming structure is affected by local acceleration of water flow due to the internal erosion development, which allows to detect leakages and internal erosion.*

I'll see where to include this in the text, whether in the introduction (Section 1) or in the conceptual model (Section 4).

*# p.8, rows 22-26 "For this reason, the mechanism that promoted the formation of piping-induced thermal anomalies was not, as commonly found in the literature, the prevalence of advection in the eroded regions in opposition to the purely conductive behaviour of the regions unaffected by piping. This occurs because the assumption of conductive behaviour in the unaffected regions (Johansson and Hellström 2001; Johansson and Sjödal 2009) only applies to low permeability bodies such as dam cores.." Remark : The thermal effect of piping development in soil, comparing with the case of soil without piping, was analyzed for different value permeability of soil and for different advection transport intensity (Péclet value from 0.1 up to 100) in the thesis of Radzicki (2009) and presented as well in Radzicki and Bonelli (2010).*

I will certainly reformulate and include this. I think it is correct to say that the first models developed for leakage detection in dams, for which numerical methods are also available to quantify the water flow, assumed the conduction was dominant in the regions unaffected by piping. Later the thermal effect of piping development in soil was analyzed for different values of soil permeability and for different advection transport intensity (Radzicki, 2009; Radzicki and Bonelli, 2010).

---

## Author Response (AR2)

Dear Reviewer and dear Editor,

we contacted the companies that provided the fibre-optic sensing system and performed the measurements. They provided information that in our opinion were sufficient to answer the questions related to the calibration of the sensor and the effect of strain on the measurements.

The following paragraphs were added in the manuscript.

Section 2.2:

"The sensing system was calibrated in the laboratory. Although field calibration is generally recommended, laboratory calibration was considered sufficient in this case, since the main interest was on temperature variations (both in space and in time) rather than on absolute temperature values. A reproducibility of less than 0.1°C was obtained for a fibre length less than 1000 m, which was the case in the field experiment. Another feature that can be source of inaccuracy in the measurements is signal attenuation due to sharp bends in the fibre. In the setup the curvature radius between each strip was more than 20 cm, which produces no attenuation. "

Section 2.3:

"It can be excluded that the temperature measurements were influenced by excessive strain experienced by the fibre during the experiment. Indeed the maximum strain measured by the single-mode fibre after four days of testing was around 400 µstrain. The strain experienced by the multimode fibres dedicated to temperature measurements was certainly much smaller since these fibres are placed into loose tubes to prevent excessive stresses."

Pictures that were not introduced in the manuscript but might be of interest for the reviewer are given in the following page.

The comments about minor issues have all been used to improve the manuscript.

[Figure]

Figure 1 - Laboratory calibration of the sensing system.

**STRAIN**

[Figure]

Figure 2 - Distributed strain measurements after four days of testing (source: Report TenCate GeoDetect on IJkdijk AIO SVT, October 15[th] 2012).

---

## Author Response (AR3)

Dear Editor,

we added in the paper two references to the suggested paper. These references indeed strengthen our statements concerning the accuracy of the sensing system. We hope this will be sufficient.

Best regards,

Silvia Bersan and coauthors

---

## Author Response (AR4)

Dear Editor and dear reviewer,

I apologize for the previous reaction to your comments. We hadn't completely understood your requests and this is the reason why our answer was so short and terse.

We took the time to study the problem more in depth and we have now a better understanding of the problem and, hopefully, also of your point of view.

The parts that we introduced/changed in the paper in order to to address the issue of the sensor accuracy are reported in the following.

I send my best regards, also on behalf of my coauthors.

Silvia Bersan
* * *
**Section 2.2 Piping test on a large-scale trial embankment**

"According to the laboratory calibration the accuracy of the measurements was higher than 0.1°C up to a fibre length of 1 km. Please note that in the field, the accuracy may be affected by additional installation issues not considered in laboratory calibration, such as sharp bends in the deployed cable, connections and splices, which may induce step losses not addressed by laboratory calibration (Tyler et al., 2009). In the field test, significant effort has been paid to avoid sharp bending: the cable has been deployed with curvature radii larger than 20 cm. Moreover, neither splices nor connectors were present in the section of fibre under measurement. Nonetheless, splices and connectors were present elsewhere, and therefore a field calibration of the DTS system would have been advisable to compensate the corresponding step losses (Hausner et al., 2011; Hausner and Kobs, 2016). Unfortunately, field calibration was not carried out due to practical issues; for this reason some impairments on the overall accuracy of the system may be expected. Such impairments are expected to be of a rather small magnitude as the temperature variations expected over time are modest."

**Section 3.2 Temperature data**

"The temperature anomalies detected were of very small extent, in the order of few tenths of a degree. […] The small extent of the anomalies could also raise concern about the impact that the accuracy of the sensor has on the results. As previously stated, according to the laboratory calibration the accuracy of the measurements was higher than 0.1°C, but in the field the accuracy may be affected by step losses occurring at sharp bends, connections and splices along the cable. Since the main interest here is in relative measurements, i.e. temperature variations in space and in time, significant influence of step losses can be excluded because neither splicings nor connectors were present in the section of fibre installed under the dike. Moreover, if we consider the most informative part of the sensor, i.e. the most downstream line F1, also step losses induced by bends can be reasonably excluded. Of course all the step losses occurring between the reading unit and the line F1 could affect the accuracy of the measurements performed at the latter, but their effect on the relative measurements is expected to be negligible. It can be also excluded that the temperature measurements were influenced by excessive strain experienced by the fibre during the experiment. Indeed, the maximum strain measured by the single-mode fibre after four days of testing was around 400 μstrain. In addition, the strain experienced by the multimode fibres dedicated to temperature measurements was certainly much smaller since these fibres are placed into loose tubes to prevent excessive stresses."